# Repeatability of evolution and genomic predictions of temperature adaptation in seed beetles

Alexandre Rêgo [1,2] ✉, Julian Baur [1], Camille Girard-Tercieux[1,3],
Maria de la Paz Celorio-Mancera [4], Rike Stelkens [2] & David Berger [1]

Climate warming is threatening biodiversity by increasing temperatures beyond the optima of many ectotherms. Owing to the inherent non-linear relationship between temperature and the rate of cellular processes, such shifts towards hot temperature are predicted to impose stronger selection compared with corresponding shifts towards cold temperature. This suggests that when adaptation to warming occurs, it should be relatively rapid and predictable. Here we tested this hypothesis from the level of single-nucleotide polymorphisms to life-history traits in the beetle *Callosobruchus maculatus*. We conducted an evolve-and-resequence experiment on three genetic backgrounds of the beetle reared at hot or cold temperature. Indeed, we find that phenotypic evolution was faster and more repeatable at hot temperature. However, at the genomic level, adaptation to heat was less repeatable when compared across genetic backgrounds. As a result, genomic predictions of phenotypic adaptation in populations exposed to hot temperature were accurate within, but not between, backgrounds. These results seem best explained by genetic redundancy and an increased importance of epistasis during adaptation to heat, and imply that the same mechanisms that exert strong selection and increase repeatability of phenotypic evolution at hot temperature reduce repeatability at the genomic level. Thus, predictions of adaptation in key phenotypes from genomic data may become increasingly difficult as climates warm.

Whether evolution is repeatable, and whether the same genes contribute to recurrent phenotypic adaptations, is a long-standing question with fundamental implications for predicting evolutionary responses to changing environments[1–3]. Recent studies have shown that, while evolution at the phenotypic level often is repeatable, the genes that contribute to adaptation can be highly contingent on the evolutionary history of populations[2,4,5]. Nevertheless, theory[6] and recent evolve-and-resequence experiments[2,7–13] have revealed that broad-scale patterns also emerge at the genomic level; adaptation is often polygenic, and while individual DNA polymorphisms are unlikely to repeatedly contribute to adaptation, the same molecular pathways are often involved, sometimes even over long evolutionary timescales[14]. This type of information is necessary to gain an informed opinion about adaptive potential under climate change[1,15,16], and is becoming increasingly important given the growing reliance on genomic data to inform conservation practices[17–20]. However, whether genomic estimates alone

[1]Department of Ecology and Genetics, Uppsala University, Uppsala, Sweden. [2]Department of Zoology, Stockholm University, Stockholm, Sweden. [3]AgroParisTech, INRAE, UMR Silva, Université de Lorraine, Nancy, France. [4]Department of Ecology, Environment, and Plant Science (DEEP), Stockholm University, Stockholm, Sweden. ✉e-mail: denovorego@gmail.com

**Fig. 1 | Experimental design. a**, Populations were collected from California (yellow), Brazil (teal) and Yemen (orange) and adapted to benign lab conditions (29 °C) for approximately 200 generations. Thermal performance curves (with 95% confidence intervals shown as shaded bands) of the three populations show the classic asymmetric shape and depict how population growth rate (adult offspring production/generation time) changes with temperature. While mean growth rates are slightly lower at 23 °C compared with 35 °C, the slope of the performance curve is steeper at temperatures higher than the optimum. To start experimental evolution, lines were placed at either 23 °C or 35 °C, with two replicates in each treatment per genetic background. Horizontal black bars in the arrows depicting each experimental evolution line indicate the time at which lines were sequenced. The sequencing of heat-adapted lines at generation 3 represent the genomic estimates of ancestral variation. **b**–**e**, For both the seven life-history phenotypes (**b**,**c**) and for all genomic SNPs (**d**,**e**), principal component analysis (PCA) summaries are presented of evolutionary change between ancestral (grey) and heat- (red) and cold-adapted (blue) lines. Phenotypes were measured in a common garden design including both the low and high temperature. The world map was generated using the R packages ggplot and maps. Scientific illustration of *C. maculatus* drawn by Milena Trabert.

can accurately predict extinction risk and adaptation in key ecological phenotypes remains poorly understood[3,21,22].

Several factors can influence the probability of observing repeated evolutionary responses, such as differences in standing genetic variation in natural populations[23–25], or previous differentiation between evolving lineages that may impact the fates of segregating alleles whose fitness effects depend on allele frequencies at other loci (that is, epistasis)[26–28]. Theory additionally suggests that environmental changes that impose stronger selection are more likely to produce repeatable evolutionary outcomes compared with environments that impose weak selection[2,6]. However, despite being critical for eco-evolutionary forecasting, the relative importance of these factors in producing (non)repeatable outcomes of evolution are not well understood[3,29,30].

Genetic adaptation to temperature is becoming evermore crucial under contemporary climate change. To what extent high and low temperatures result in repeatable evolutionary outcomes, and whether changes at the genomic level are informative of phenotypic adaptation, can further inform the future management of threatened populations. However, studies that incorporate both phenotypic and genomic information on thermal adaptation are rare[31,32]. Yet, cases of adaptation to temperature are amenable to exploring the causes of evolutionary repeatability due to a fairly deep understanding of thermal physiology[33,34]. For example, thermodynamic constraints on cellular processes often manifest as asymmetries in thermal performance curves of ectotherms, where performance measures more rapidly decline at hotter than colder temperatures at equal distances from a thermal optimum[35] (Fig. 1a). Organisms far-displaced from their thermal optimum towards hotter temperatures may thus experience strong increases in selection for beneficial alleles, suggesting that evolution may generally be faster and more predictable in warming climates[36–38]. Understanding the evolutionary consequences of this physiological

temperature dependence is thus of central importance for evaluating the prospects of employing genomics data to predict phenotypic adaptation and extinction risk under future climate change.

Here we combine data on phenotypic and genomic change from replicate experimental evolution lines of the seed beetle *Callosobruchus maculatus*, deriving from an evolve-and-resequence experiment designed to quantify the role of selective determinism versus historical contingency in thermal adaptation. Replicate lines, established from three far-separated geographic populations, were placed at cold (23 °C) or hot (35 °C) temperature equidistant from their ancestral temperature (29 °C; Fig. 1a). To test for evolutionary repeatability, we estimated evolutionary trajectories at the phenotypic level among seven life-history traits, and at the genomic level across putatively selected single-nucleotide polymorphisms (SNPs) and genes. We then compared evolutionary responses between line replicates (within genetic backgrounds) and geographic populations (between genetic backgrounds) to understand the role of historical contingency and temperature-specific selection on the repeatability of evolution. Finally, we estimated the correspondence between genomic and phenotypic estimates to evaluate the power of genomic data to predict the level of (mal)adaptation.

## Results

### Phenotypic evolution is faster and more parallel at hot temperature

We first re-analysed data on thermal adaptation in seven life-history traits of females[39], including three core traits: lifetime offspring production (LRS), adult weight and juvenile development time (Fig. 2a); as well as four rate-dependent traits measured over the first 24 hours of female reproductive lifespan: weight loss, water loss, early fecundity and mass-specific metabolic rate (Fig. 1b,c and Extended Data Fig. 1).

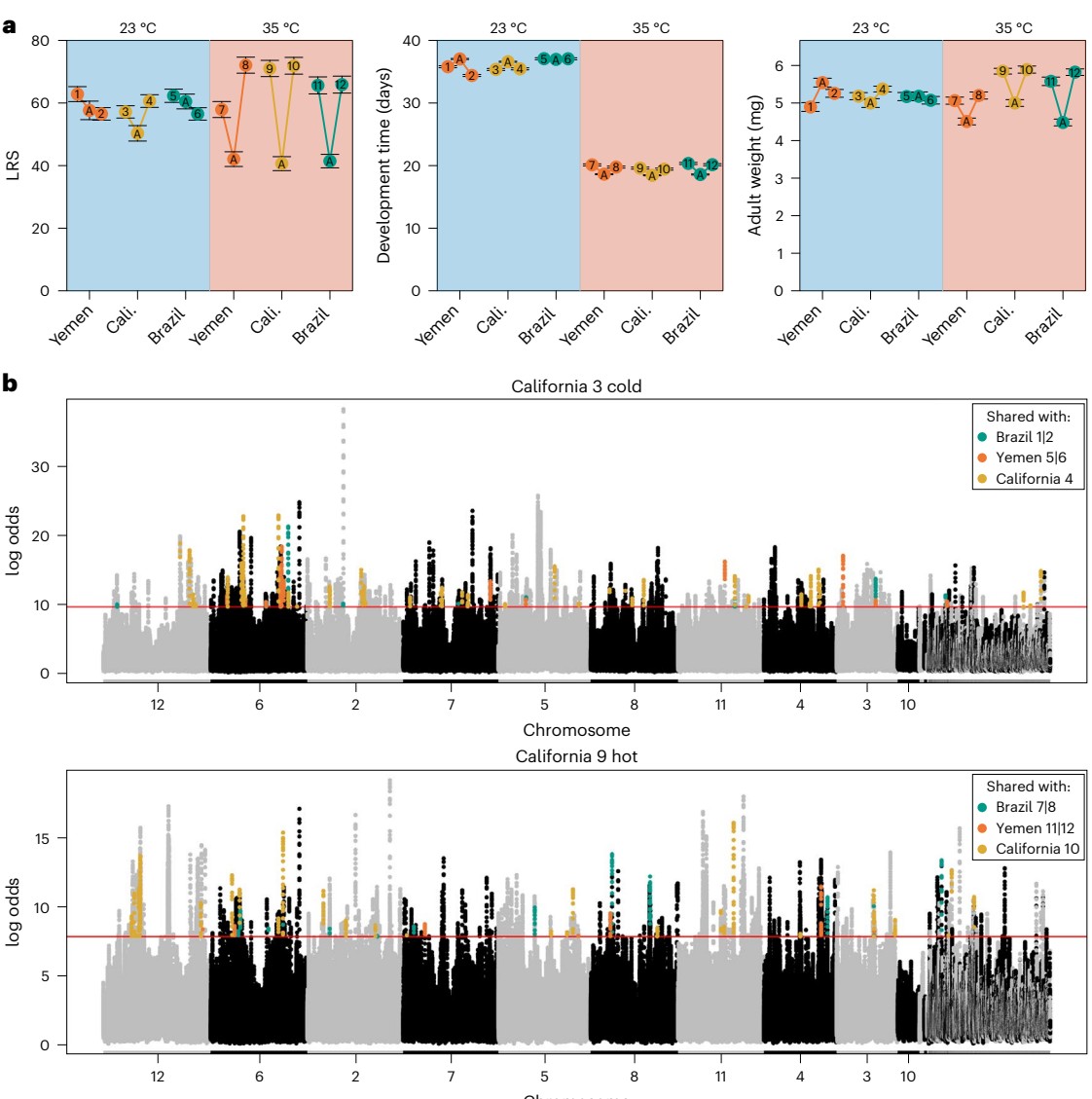

**Fig. 2 | Phenotypic evolution and example of genomic data. a**, Means (±1 s.e.m.) are shown for ancestors (A) and evolved lines for three of the seven measured traits (see Extended Data Fig. 1 for all traits). Phenotypes of evolved lines (1–12) are shown only as measured in their local temperature regime (cold lines, 1–6, light blue shading, measured at 23 °C; hot lines, 7–12, light red shading, measured at 35 °C). Cali., California. **b**, Manhattan plots of rolling average log odds *P* values (window size = 20 SNPs) for one heat-adapted and one cold-adapted line from the California background (see Supplementary Fig. 1 for all populations) across all putatively selected sites (*n* = 475,194) using R package RcppRoll (v.0.3.0). The largest ten scaffolds (that is, chromosomes) are labelled. The 0.001th quantile of rolling average *P* values is given by the horizontal orange line for each line. SNPs whose rolling average *P* values are also above their respective 0.001th quantile in other populations are coloured. There is a clear polygenic signal of adaptation. Note that the California lines shown here exhibit many more SNPs in common with the single other California replicate (yellow points) than those shared with any of the four line replicates of different origin (teal and orange points).

By comparing each of the 12 evolution lines with their respective ancestor, we assessed the direction and magnitude of phenotypic evolution at 23 °C (comparing ancestors with cold lines) and 35 °C (comparing ancestors with hot lines). We quantified the repeatability of this evolution at each temperature as geometric angles between evolutionary change vectors in multivariate trait space, where an angle of 90° signifies uncorrelated responses and angles of 0° and 180° correspond to perfectly parallel and anti-parallel responses, respectively. Comparisons were made between pairs of evolution lines deriving either from the same or different genetic backgrounds. We also quantified whether populations were converging or diverging over the course of the experiment by comparing the distance between populations at the start ($S_d$) of experimental evolution (that is, between ancestors) to the distance at the end ($E_d$; that is, between the evolved lines).

Hot lines exhibit higher evolutionary rates per generation ($\| \bar{x} \| = 0.87 \pm 0.14$, mean ± s.d.) compared with cold lines ($\| \bar{x} \| = 0.5 \pm 0.07$) (t-test$_{d.f.}$, $t_5 = -4.01$, $P = 0.003$; Fig. 3a). Phenotypic changes were also more parallel ($\theta \pm$ s.d.) on average (permutation test, $P < 0.001$) in hot lines (39.32 ± 19.16) than cold lines (67.42 ± 23.3), consistent with the hypothesis that high temperatures impose stronger selection (Fig. 3b). Compared with Monte Carlo simulations of random angles in seven-dimensional space, the mean pairwise angles for both cold and hot lines were more parallel than expected by chance (two-sided test, $P < 0.05$, $10^5$ iterations). Moreover, parallelism was greater within than between genetic backgrounds for both hot and cold lines (Supplementary Table 1), suggesting a role of historical contingency in dictating the repeatability of phenotypic evolution.

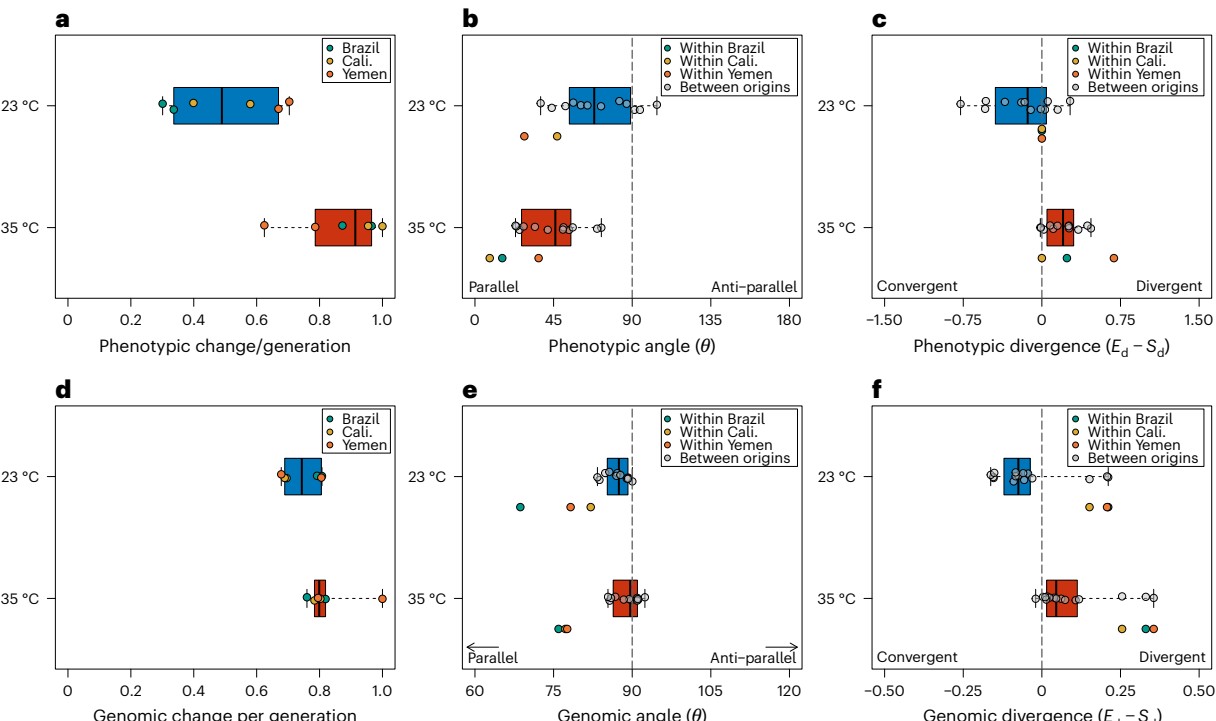

**Fig. 3 | Evolutionary rates, pairwise angles and measurements of divergent evolution. a–f,** Distribution of all phenotypic and genomic evolutionary rates (**a**,**d**), pairwise angles (**b**,**e**) and measurements of convergent or divergent evolution (**c**,**f**) for cold (blue) and hot (red) lines are shown. Boxplots display the median (solid centre line), interquartile range (bounds of the box) and range (whiskers) of the data. All genomic estimates are based on the set of SNPs putatively under selection in any population ($n = 475,194$). Angles and divergences are given for pairwise comparisons between lines of different (grey points) and same (coloured points) genetic background. Angles of 0° represent completely parallel responses, while angles of 90° represent uncorrelated responses. Divergence was calculated as the difference in Euclidean distance between population pairs at the end ($E_d$) and start ($S_d$) of experimental evolution, and are divergent when the distance between end points is greater than the distance between start points ($E_d − S_d > 0$), or convergent if the opposite is true ($E_d − S_d < 0$). Evolutionary rates have been scaled by the maximum evolutionary rate in the dataset, while divergences have been scaled relative to the distance between the two most differentiated ancestors. Inestimable phenotypic angles due to sampling error are not shown (within Brazil, 23 °C).

Despite increased parallelism during adaptation to heat, cold lines generally exhibited less divergence (Div; or greater convergence) relative to hot lines (Fig. 3b; $Div_{23} = −0.14 ± 0.29$; $Div_{35} = 0.22 ± 0.20$, permutation test, $P < 0.001$). While this result, to some extent, can be explained by the overall smaller magnitudes of per-generation phenotypic change in cold relative to hot lines, it also suggests that non-parallelism among cold lines in part resulted from convergent evolutionary trajectories of differentiated ancestral populations, making cold lineages become more similar to each other in some phenotypic traits during experimental evolution (Fig. 2a).

### A polygenic basis of thermal adaptation

We estimated allele frequency shifts and selection coefficients using whole-genome sequences (pool-seq) from all 12 evolved populations and their 6 ancestors (Fig. 1a). Consistent with stronger selection at warmer temperature, hot lines experienced greater rates of allele frequency change at candidate loci and were estimated to have lower effective population size ($N_e$) than cold lines (Supplementary Table 2). This pattern was similar for selection coefficients that take into account drift based on estimates of $N_e$, with hot lines experiencing stronger average selection on all three backgrounds (Supplementary Table 2).

Thermal adaptation was highly polygenic, involving several thousands of candidate SNPs (Supplementary Table 3). Although it is unlikely that these SNPs are causal, several distinct peaks can be observed of SNPs deviating from drift expectations across almost all chromosomes in all populations (Fig. 2b and Supplementary Fig. 1). We identified SNPs under putative selection on each genetic background separately and assigned these SNPs into four categories: synergistically pleiotropic—SNPs selected in the same direction across both thermal regimes; antagonistically pleiotropic—SNPs selected in opposite directions across regimes; and private cold and private hot—SNPs selected in only the cold or hot regime, respectively (Fig. 4a). Contrary to typical theoretical assumptions, overall patterns indicate that thermal adaptation is not governed by antagonistic pleiotropy. Instead, thermal adaptation was primarily characterized by privately selected alleles. There were also more SNPs evolving in the same direction between temperature regimes than in opposite direction. These general patterns seem not affected by potential detection bias against pleiotropic, relative to private, SNPs (Supplementary Fig. 2). Compared with forward-in-time simulations from ancestral populations (R package poolSeq v.0.3.5.9), there were more SNPs shared between the two line replicates within each background than expected under drift (all $P < 0.001$). However, most candidate SNPs were unique to a particular genetic background, with only private alleles selected at cold temperature showing modest repeatability across backgrounds (Extended Data Fig. 2).

As expected, there was more repeatability at the level of genes (Fig. 4b), with an excess of shared variants among all pairwise comparisons across origins (all $P < 0.01$). Across all six cold lines, there were 296 shared genes targeted by selection (that is, containing a putatively selected SNP), which is many more than expected by chance (expected = 2.33, $P < 0.001$), while hot lines shared 51 genic targets (expected = 0.11, $P < 0.001$; see Supplementary Table 5 for associated Gene Ontology (GO) terms). Repeatability was again more pronounced for privately selected genes and at cold temperature. Jaccard indices of the overlap of candidate genes indicated that cold lines show greater repeatability ($0.33 ± 0.06$) compared with hot lines ($0.21 ± 0.05$; permutation test, $P <$

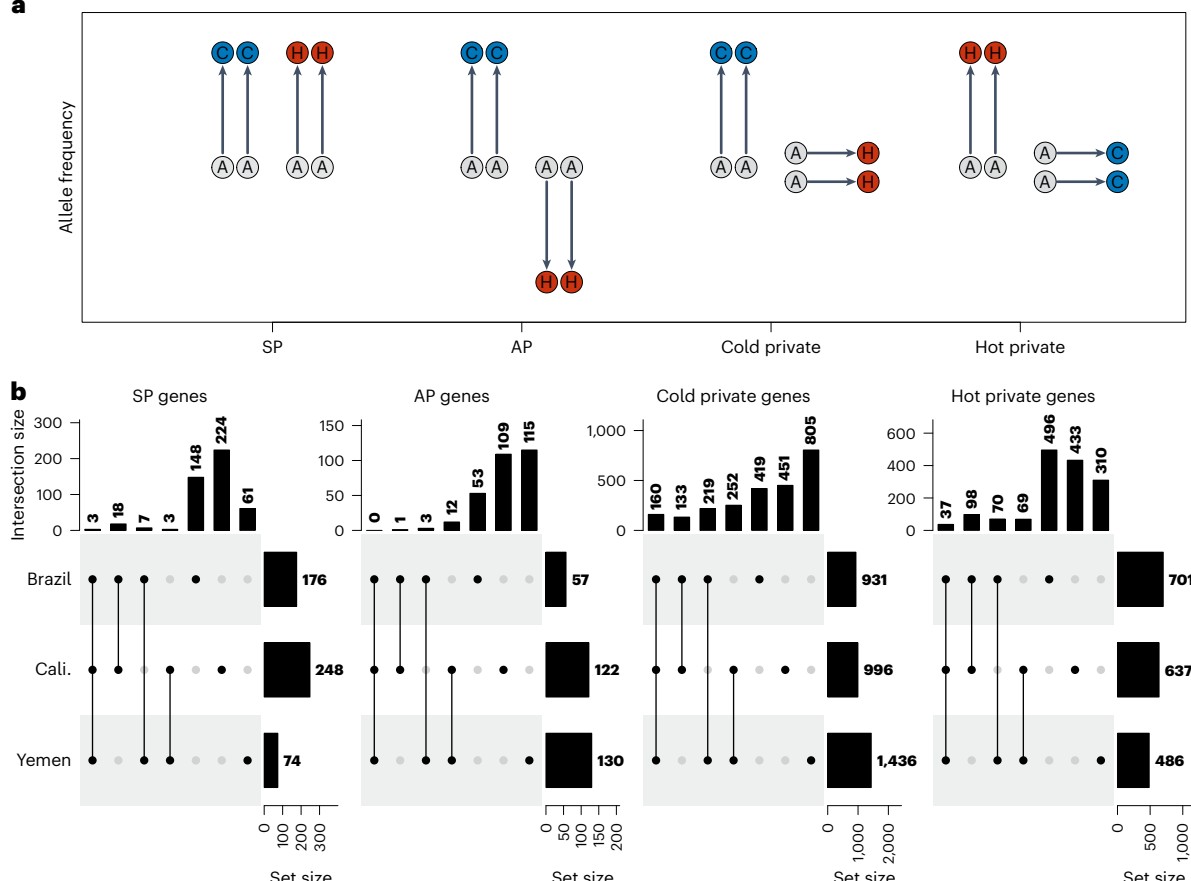

**Fig. 4 | The criteria for allele classifications and overlapping gene sets of those alleles. a**, Schematic showing how SNP and gene classes were assigned based on patterns of allele frequency change from the ancestor (A) to the evolved cold (C) and hot (H) lines. For each class, we require both replicates of a particular background to behave similarly (that is, more extreme than the 1st or 99th percentiles with the same direction of allele frequency change in both replicates). The four classes identified are: (1) synergistically pleiotropic (SP) SNPs, which are selected for in the same direction between regimes; (2) antagonistically pleiotropic (AP) SNPs, which are selected in opposite directions between regimes; (3) private cold SNPs, which are selected for only in the cold regime; and (4) private hot SNPs, which are selected for only in the hot regime. **b**, Upset plots of gene sets based on corresponding SNP set classes (Extended Data Fig. 2) located in protein-coding regions. There is significant excess of shared private selection genes, and more gene sharing among genetic backgrounds adapting to cold temperature.

0.001; Supplementary Fig. 3). We performed detailed scans across the genome for outlier windows showing recurrent allele frequency change among replicate lines using the R package AF-vapeR. This revealed that cold adaptation was characterized by regions that were selected upon across nearly all replicate lines (Extended Data Fig. 3a), while heat adaptation was more often characterized by background-specific genomic responses (Extended Data Fig. 3b). Hence, thermal adaptation seems best described by polygenic adaptation with effects private to each temperature, with modest repeatability across genetic backgrounds for heat adaptation but higher repeatability when evolution proceeded at cold temperature (Fig. 4b).

Although no antagonistically pleiotropic genes were shared between all backgrounds, such genes shared between two origins contained a cGMP-dependent protein kinase, which is associated with thermotolerance in *Drosophila melanogaster* larvae[40], and the gene *couch potato* (*cpo*), which is associated with diapause climate adaptation in both *D. melanogaster*[41] and *Culex pipiens*[42]. Gene set enrichment analysis (Extended Data Fig. 4) revealed that both synergistically and antagonistically pleiotropic genes were over-represented for processes such as oxidation–reduction, double-strand break repair and DNA recombination, which may indicate general responses to suboptimal temperature[33]. The genetic backgrounds shared many more processes for privately selected gene sets. Notably, the GO term

'Homophilic cell-adhesion via plasma membrane adhesion molecules (GO:0007156)' is found in almost all gene set categories, across all backgrounds. This GO term was also identified in several *Anolis* species in relation to adaptation of thermal niches[43] and cold tolerance in *D. melanogaster*[44]. As the mechanical and functional properties of the plasma membrane are greatly influenced by thermal fluctuations[45], it seems likely that this biological process is a strong candidate for shared targets of selection across thermal gradients.

### Genomic evolution is more predictable at cold temperature

We estimated the repeatability of genomic adaptations taking the same approach as for the phenotypic data and quantified angles between vectors of evolutionary change (here represented by shifts in allele frequencies) between lines deriving from the same or different genetic backgrounds. We aimed to ascribe observed temperature-specific patterns to three key factors expected to impact the predictability of evolution: (1) environmental differences in the strength of selection (expecting more repeatability at hot temperature); (2) differences in available ancestral genetic polymorphisms at selected genes; and (3) genetic epistasis between selected genes and differentially fixed loci in the three ancestors. To achieve this, we first conducted analyses using all SNPs to detect candidate loci under selection on each of the three genetic backgrounds. We then repeated the analyses using only

those SNPs that were shared among the backgrounds, expecting our estimates of repeatability to increase if differences in ancestral polymorphisms were key in driving the patterns.

Despite greater phenotypic parallelism in response to heat, hot lines did not exhibit greater parallelism at the genome level ($\bar{\theta}_{23} = 83.42 \pm 6.7$, $\bar{\theta}_{35} = 85.48 \pm 7.8$, permutation test, $P = 0.15$; Fig. 3c). Note that relatively orthogonal measures of parallelism compared with phenotypic estimates are ascribed to the difference in dimensionality of the two analyses. Indeed, the observed pairwise angles among both cold- and heat-adapted lines were much smaller than expected by chance (Monte Carlo simulations, $P < 0.001$, $10^5$ iterations; Extended Data Fig. 5). Just as for the estimates of phenotypic repeatability, within-background comparisons were more parallel ($\bar{\theta}_{23} = 72.96 \pm 7.16$; $\bar{\theta}_{35} = 72.28 \pm 1.5$) than between-background comparisons, and especially so for heat adaptation ($\bar{\theta}_{23} = 86.03 \pm 3.25$; $\bar{\theta}_{35} = 88.77 \pm 4.19$), suggesting an important role of historical contingency in dictating genomic repeatability (Supplementary Table 4). We note that these qualitative patterns hold true even when estimating parallelism from temperature-specific sets of SNPs with differing dimensionality and are unlikely to be due to differences in the number of selected sites between regimes (Extended Data Figs. 5a,b and 6c,d).

Cold lines tended to diverge less (or converge more) in genomic space than hot lines (permutation test, $P = 0.011$). This pattern is true both within (Div$_{23}$ = $0.22 \pm 0.03$; Div$_{35}$ = $0.32 \pm 0.06$) and between backgrounds (Div$_{23}$ = $-0.13 \pm 0.04$; Div$_{35}$ = $-0.01 \pm 0.06$; Fig. 3f). Unlike phenotypic rates, genomic rates of change at selected loci were not different between hot and cold lines ($t_5 = -0.59$, $P = 0.57$; Fig. 3d).

That we see greater phenotypic repeatability for hot lines, but higher genomic repeatability across cold lines, for all genomic metrics is counter-intuitive and opposite of our expectations. We propose three non-mutually exclusive mechanisms that may contribute to such patterns. First, owing to stronger selection at high temperature, hot lines also experience more genetic drift, as evidenced by lower $N_e$. This could contribute to lower predictability of genomic changes across lines. However, $N_e$ was not strongly correlated with the repeatability of observed evolutionary trajectories (Supplementary Fig. 4a,b) nor with divergence at candidate loci within regimes (Supplementary Fig. 4c,d).

Second, differences in ancestral polymorphism at selected loci may contribute to historical contingencies and low repeatability across geographically separated populations[23]. To test this hypothesis, we re-estimated all measures of genomic repeatability based on only those putatively selected SNPs that segregated in all ancestral populations ($n = 119{,}630$, or ~25% of the original set), expecting to see substantially higher repeatability, especially for heat-adapted lines, if differences in ancestral polymorphism have contributed to the observed temperature-specific patterns. However, although genomic repeatability was slightly increased overall for shared polymorphic sites, the greater effect of genetic background on parallelism and divergence observed for heat relative to cold adaptation remained (Supplementary Fig. 5, Extended Data Fig. 6 and Supplementary Table 6). Additionally, when comparing the overlap between backgrounds of candidate genes in the privately selected gene sets (which contain the majority of candidate SNPs), we found no difference with respect to using all SNPs or only those shared among all backgrounds (Supplementary Fig. 6, all chi-squared test $P > 0.05$). Hence, differences in shared ancestral polymorphisms seem to have played a minor role in shaping the observed genomic repeatability between temperatures.

Third, it is well recognized that gene interactions (epistasis) can contribute to unpredictable evolutionary outcomes[46], and that such epistasis might depend on the environment[27,47]. Indeed, epistasis is predicted to be a natural property of metabolic networks, and as all metabolic processes are strongly temperature dependent, it is possible that epistasis between differentially fixed variants in the three genetic backgrounds and selected variants may have been stronger at hot temperature. In Fig. 5 and Supplementary Fig. 11, we illustrate

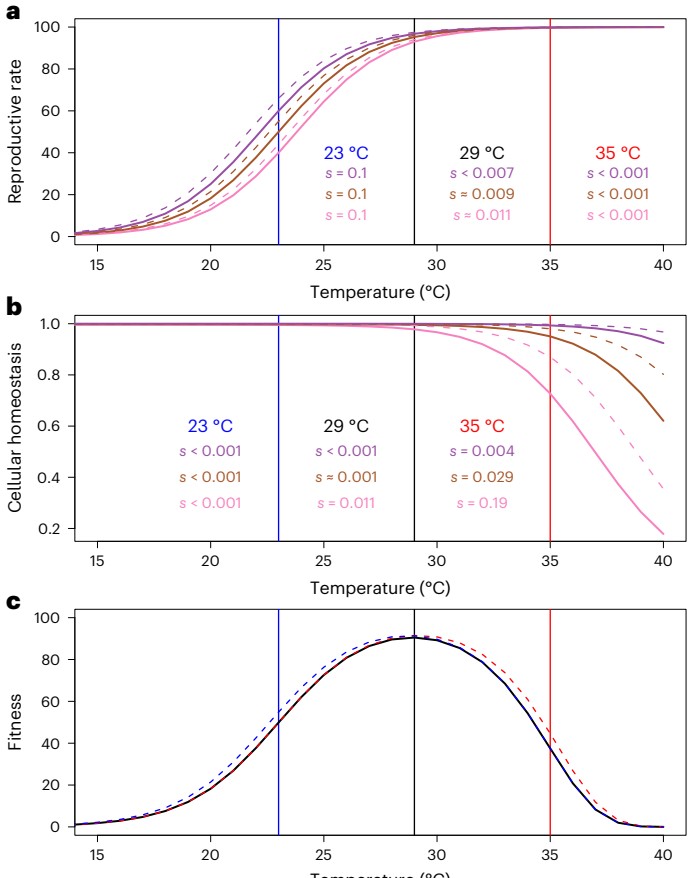

**Fig. 5 | A prediction of temperature-dependent epistasis for fitness based on thermodynamics of cellular processes.** For each plot, coloured vertical lines denote the temperature of our thermal regimes (23 °C and 35 °C). **a,** Reproductive rate increases exponentially with temperature at the colder range due to relaxation of thermodynamics constraints on enzymatic reactions, but starts to follow a pattern of diminishing returns at warmer temperatures due to ecological constraints placing limits on reproductive output. Illustrated for three hypothetical genetic backgrounds (purple, brown and pink) with wild types (solid lines) and a mutant (dashed lines, whose selection coefficient is denoted by $s$) with a 10% increase in enzymatic reaction rate at 23 °C. This results in strong selection on the mutation at cold temperature for all backgrounds as these are far from their optimal reproductive rate, but weak selection at hot temperature at which all backgrounds are close to the maximal achievable reproductive rate determined by ecological factors. Epistasis is weak and selection is similar across backgrounds at all temperatures. **b,** Warm temperatures increase a range of molecular failure rates, which results in less enzyme ready to catalyse reactions, and more misfolded proteins within cells, leading to protein toxicity and depressed fitness. Illustrated for three genetic backgrounds with stable (purple), intermediate (brown) and unstable (pink) wild-type protein. A mutation increasing stability (dashed lines) was introduced on each background. Selection on the mutation is weak at cold temperature for all backgrounds but can become very strong at hot temperatures depending on the wild-type protein stability, and strong epistasis for fitness results. **c,** Fitness, as the product of reproductive rate and molecular failure rate, is depicted for the wild type (black), and the mutants with increased enzymatic reaction rate (blue) and increased protein stability (red) for the pink background.

that our temperature-specific estimates of genomic adaptation (that is, excess of privately selected SNPs) and repeatability (lower at hot temperature) are consistent with predictions from a simple model of thermal adaptation assuming exponential increases in both enzyme reaction rates (Fig. 5a) and molecular failure rates (for example, loss of enzyme stability; Fig. 5b) with increasing temperature[37,48]. This heuristic model has been used previously to argue that hot temperature

results in increased epistasis by causing proteins to evolve marginal thermostability, such that further mutational or thermal perturbations can be either inconsequential or highly deleterious depending on the wild-type protein[37,38,49–51]. Nevertheless, the mechanistic basis underlying the observed temperature specificity of evolutionary repeatability needs to be investigated further.

### Predicting phenotypic (mal)adaptation from genomic evolution

To assess the premise of predicting phenotypic adaptation from genomic data alone, we calculated 'offsets' for both types of data, describing the level of genetic and phenotypic maladaptation relative to the most well-adapted line, which we used as the refs. [52,53]. We performed this calculation for each of the three backgrounds and for hot and cold assay temperature separately. We here focus on offsets at hot assay temperature as we see more clear signs of local adaptation to heat than cold, and because phenotypic estimates of cold adaptation were smaller and many times not statistically significant (Extended Data Fig. 1). For completeness, we present the results for cold assay temperature in Supplementary Figs. 7 and 8.

Genomic offsets were strongly predictive of phenotypic distance ($r_{Pheno}$) and fitness offsets ($r_{Fit}$) among populations reared at hot temperature for lines of the same geographic origin as the reference ($\bar{r}_{Pheno} = 0.88$, analysis of covariance (ANCOVA) $F_{1,9} = 13.6$, $P = 0.005$; $\bar{r}_{Fit} = -0.74$, ANCOVA $F_{1,9} = 7.23$, $P = 0.025$). However, when applied to lines of different geographic origin, the predictive power of genomic data was severely reduced ($\bar{r}_{Pheno} = 0.22$, ANCOVA $F_{1,30} = 0.86$, $P = 0.36$; $\bar{r}_{Fit} = -0.29$, ANCOVA $F_{1,30} = 1.24$, $P = 0.27$; Fig. 6). These results remained qualitatively unchanged when considering only SNPs that were polymorphic in all ancestors (Supplementary Fig. 9), suggesting that differences in ancestral polymorphisms were not per se causing the poor predictability across genetic backgrounds. Hence, selected sites identified in a particular origin do not contribute greatly to phenotypic or fitness variation in other origins, which is congruent with estimates of both phenotypic and genomic repeatability being greater within than between origins (Fig. 3).

Interestingly, the genomic offsets were typically larger between lines of the same genetic background compared with offsets between backgrounds. This suggests that, while selected sites identified in a given reference line contribute to fitness/phenotypic variation among lines of the same geographic origin, these sites may be mostly fixed (in the direction of the reference line) in lines of the other origins. It could be, then, that by focusing on SNPs that show signs of selection in a limited set of study populations, the importance of already-fixed sites that cannot contribute towards genomic offsets may be greatly underestimated in genomic prediction of geographically distant populations. To explore if this possibility contributed to the poor genomic predictions of phenotypic adaptation applied across origins, we estimated the correlation between fitness offsets and genomic offsets, basing the latter on all candidate SNPs per thermal regime (that is, identified in any of the six line replicates), with the reasoning that this should better capture and include differentially fixed sites. However, even when all putatively selected sites were considered, predictive power remained high in the case of within-origin estimates ($r = -0.87$), and low for between-origin estimates ($r = 0.32$; Supplementary Fig. 10b). Compared with offsets utilizing selected SNPs, sets of randomly chosen SNPs performed worse for within-origin estimates, but on par with across-origin predictions (Extended Data Fig. 7).

## Discussion

Climate change is disrupting ecological niches and changing species-distribution patterns across the globe[54,55]. For many taxa, the potential to genetically adapt to these changes is necessary to avoid extinction. Approaches harnessing genomic data to predict adaptive potential in key ecological traits of threatened populations may provide means to focus conservation efforts and are becoming more common, but their limitations are debated[15,17,18,56,57]. Here we adapted seed beetles from three different geographic origins to cold and hot temperature and assessed both life-history and genomic adaptation to (1) determine the genetic basis of thermal adaptation; (2) quantify the repeatability of evolution across biological levels of organization; and (3) evaluate the prospects of genomic prediction of adaptive potential across geographically isolated populations. We show that adaptation to both cold and hot temperature is highly polygenic. Evolutionary responses were more repeatable within, than between, genetic backgrounds, demonstrating an important role of historical contingency in dictating evolutionary predictability. Heat adaptation resulted in greater phenotypic rates of change, indicative of high evolutionary potential under future climate warming. However, despite a high repeatability of phenotypic evolution at hot temperature, adaptation at the genomic level was more repeatable at cold temperature, with the effect of genetic background playing a larger role for heat adaptation. Indeed, while genomic offsets for heat adaptation predicted maladaptation in lines from the same geographic origin reasonably well, they failed to predict maladaptation in lines from different origins. This suggests that the prospects of using data on genomic–phenotype correlations to predict more broad-scale geographic patterns of (mal)adaptation and evolutionary potential under future climate warming may be limited. Complementary hypotheses exist to explain the observed patterns, which we discuss below.

### Genetic basis of thermal adaptation

Thermal adaptation was found to be highly polygenic across all lines and regimes, with several thousand polymorphisms under putative selection, which is predicted to result in high genetic redundancy and low repeatability of adaptation at the genomic level[13,58]. Our results indicate that thermal adaptation is largely characterized by alleles with effects limited to a specific thermal range, suggesting that evolutionary potential under climate change may be fuelled by alleles with conditional fitness effects that are relatively abundant at ancestral and benign thermal conditions but become exposed to selection at thermal extremes. This notion also corresponds well with predictions of temperature-dependent selection on enzymes based on the thermodynamic basis of protein function[37,59,60] (Fig. 5). Our results also mirror experimental evolution studies in both *D. melanogaster*[61] and *D. simulans*[25,62] in which adaptation was polygenic and few targets of selection overlapped between opposing thermal selection regimes.

A possible exception to this main pattern is the observed divergence along one genomic dimension, explaining roughly 6% of all the variation at selected sites, that occurred in opposing directions for hot and cold lines (Fig. 1e, genomic PC3). This is in contradiction to our analyses of single SNPs and genes, where very few sites showed signs of antagonistic pleiotropy. It is well recognized that polygenic adaptation can result in genetic redundancy with stochastic and modest allele frequency changes at single sites, where allelic effects often show epistasis for fitness during the process of adaptation, causing a highly unpredictable genomic signal at the level of single SNPs[13,63]. While this points to the general problem of detecting polygenic selection at weakly selected loci, our results also highlight that overall genomic signals, such as that depicted along PC3, can still be picked up and be informative of the basis of adaptation, given sufficient experimental replication.

### (Non)parallel adaptation to thermal stress

Evolutionary trajectories were generally more parallel for comparisons within, than between, backgrounds for both phenotypic and genomic adaptation, demonstrating an important role of historical contingencies in shaping evolutionary outcomes. Similar results were found in studies of temperature adaptation in different founders of *D. simulans*[25,62]. Likewise, studies of wild populations have found

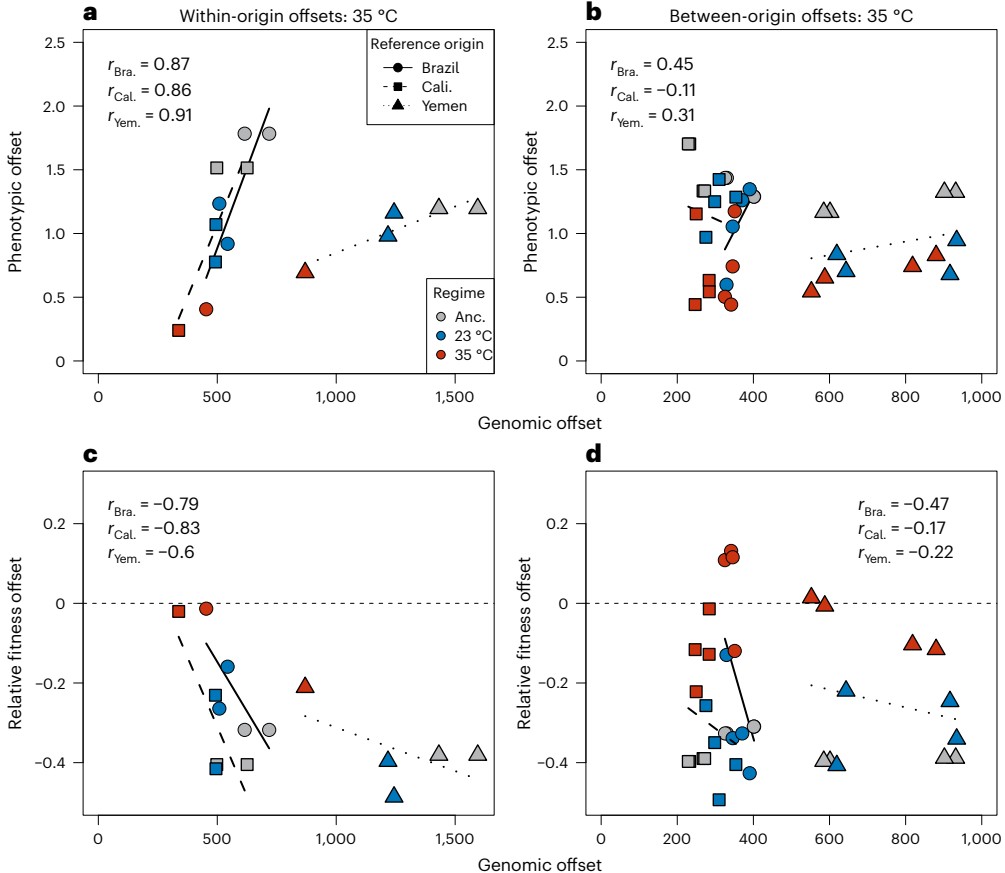

**Fig. 6 | Genomic predictors of phenotypic divergence and laboratory fitness at hot temperature (35 °C). a–d,** Genomic and phenotypic offsets were calculated using reference populations identified separately for each geographic origin. Within each origin, the line with the highest lifetime adult offspring production per generation (i.e. laboratory fitness) was designated as the reference population. Genomic offsets were calculated based on SNPs whose *P* values for allele frequency change fell within the top 0.001th quantile for each reference ($n = 10,546–10,649$). Relative fitness offsets were calculated as a line's lifetime adult offspring production per generation time, relative to that of the reference population. Similarly, phenotypic divergence offsets were calculated as the Euclidean distance in scaled trait space to this reference population. Offsets are organized by predictions within genetic backgrounds (**a,c**) and between genetic backgrounds (**b,d**). Regression lines are dotted by the origin of the reference population, while symbol colours refer to ancestral (grey), cold (blue) and hot (red) lines. Genomic offsets predict maladaptation within **a** and **c** but not between **b** and **d** geographic origins.

significant, but modest, overlap between climate adaptation loci among independently evolving lineages[64–67].

Biophysical models of protein stability predict that higher temperatures should generate stronger purifying selection[37,38,60] and increased positive selection on alleles that buffer thermal stress[33,68]. We therefore expected both faster and more repeatable adaptive evolution at hot temperature. We found evidence at the genomic level that warm temperature indeed imposed stronger directional selection based on estimated selection coefficients (Supplementary Table 2). However, contrary to our expectations, we observed that heat adaptation was only more parallel for phenotypes. Our follow-up analyses did not provide evidence for genetic drift being a causal explanation for the lack of genomic repeatability. Likewise, our estimates of repeatability based only on those SNPs that were shared between all ancestral populations suggest that the observed patterns are unlikely to be solely explained by differences in the access to (adaptive) standing genetic variation[23]. As a final explanation, epistasis between selected SNPs and differentially fixed variants in the three ancestors may have been magnified at hot temperature. Epistatic interactions between different proteins or between different SNPs within the same protein-coding sequence are ubiquitous[27,69,70], and temperature-dependent increases in such interactions are emergent properties of the assumptions in simple biophysical models of protein thermodynamics and function[37,51,60] (Fig. 5) and have been observed in empirical studies[71]. Although we

cannot explicitly test this hypothesis with the current data, the low explanatory power of both genetic drift and differences in ancestral polymorphism, together with the larger difference in repeatability between the hot and cold regime observed across (compared to within) genetic backgrounds, suggest that temperature-dependent epistasis may be causally linked to the lower genomic repeatability observed at hot temperature. This suggests, somewhat paradoxically, that the same thermodynamic mechanism that causes hot temperatures to impose strong selection, resulting in repeatability of adaptive outcomes at the phenotype level, may at the same time make genomic evolution more unpredictable by increasing epistatic interactions.

### Predicting (mal)adaptation from genomic data

Leveraging genomic datasets to predict the fates of threatened populations is an important aim in ecological and conservation genetics. However, both genetic redundancy[25] and epistasis[72] obfuscate the genotype-to-phenotype map, potentially reducing the power of using genomic data to make inferences about adaptive potential. In accordance with this notion, our genomic offsets were predictive of phenotypic adaptation to heat when applied within a single genetic background, but poor predictors when applied across genetic backgrounds. This suggests that predictions of adaptive potential under future climate change may be problematic when applied on large geographic scales, maybe to the extent that estimates of (neutral)

genome-wide genetic diversity may offer similar accuracy in predictions. Indeed, a study comparing several methods for conducting genomic offsets recently found that sets of randomly selected SNPs can sometimes better predict (mal)adaptation than identified candidate SNPs[73]. However, on the timescale of our controlled experiment, genomic offsets based on randomly selected SNPs show weaker correlations with phenotypic/fitness offsets compared with those based on candidate SNPs (especially when applied within genetic backgrounds). Moreover, we show that using randomly selected SNPs can even reverse the sign of offset correlations, resulting in highly misleading predictions (Extended Data Fig. 7).

A possible explanation for low predictability between genomic and phenotypic data is that many changes in allele frequency may not be causal for changes in phenotype. Our seed beetle populations experienced low effective population sizes ($\bar{N}_e$ = 215.4) and strong selection, which probably resulted in relatively large allele frequency changes due to genetic drift and draft, which may have resulted in skewed associations between genomic and phenotypic change. Moreover, the relatively low census population sizes (300–400 individuals) during laboratory maintenance will have contributed to further fixation of alternative alleles in the three ancestral populations, amplifying differences between the ancestors. Nevertheless, the population sizes studied here are probably representative of many threatened species, which are the prime targets for genomic prediction. Moreover, our conclusions on temperature-specific repeatability of genomic changes within and between genetic background are unlikely to be affected by low $N_e$, as our analyses focused on alleles under strong selection ($s >$ 0.01), and remained qualitatively unchanged when only considering genetic variants shared among all three backgrounds.

The particular metrics used for phenotypic and performance measures should also be given consideration[52,53]. While whole-genome analyses of predictability can typically incorporate all individual 'units' that compose the response to selection (that is, changes in allele frequency across the genome), phenotypic or fitness analyses can be sensitive to the set of traits measured[74]. Predictability should be highest at the level of fitness, relative to underlying traits[75]. The suite of traits we measured is expected to be highly associated with fitness during temperature adaptation, and should thus exhibit high predictability in evolutionary changes. Some traits chosen here have also shown strong parallel responses in *Drosophila*[25]. Nevertheless, we note that our assays did not include juvenile competition traits, which may have been central in driving cold adaptation[76], potentially explaining the lack of identified local adaptation and poor correlations between genomic and fitness offsets at 23 °C.

## Conclusions

Our work combines evolutionary change at the phenotypic and genomic level, something so far rarely done in thermal adaptation studies[31,32,62,77]. While phenotypic evolution was more rapid and repeatable at warm temperature, the opposite was true for genomic change, with historical contingencies obfuscating the relationship between genomic and phenotypic adaptation to heat. Multiple non-mutually exclusive mechanisms may have contributed to this pattern. Importantly, theory predicts that many of the same mechanisms that should increase predictability at high temperatures (for example, strong selection on cellular homeostasis) may contrarily result in reduced genomic predictability due to the biophysical properties of cells, resulting in epistasis, and the demographic properties of populations, resulting in genetic drift/draft. Our findings suggest that predicting adaptation to future warming climates using genomic data may become increasingly difficult, especially when extrapolating across geographically isolated populations, thus placing emphasis on sampling many populations to more confidently describe hypotheses regarding broad patterns of evolutionary repeatability[78]. For genomic predictions, there is a scarcity of experimental evolution studies to validate and refine predictive methods, especially with regards to the role of epistasis in dictating the relevance of genomic predictions across larger geographic scales. Such studies will be invaluable for advancing predictive accuracy of forecasts of threatened species.

## Methods

### Study species

The seed beetle, *C. maculatus*, infests human stores of grain legumes. Females attach eggs to the surface of seeds and larvae burrow into and develop within a single seed. Adults then emerge and reproduce, typically within 3–4 weeks, without the need for additional nutritional resources and a full life cycle can be completed within less than a month at benign temperatures; egg-to-adult development time is thus a good approximation of generation time in laboratory conditions. Because *C. maculatus* has been associated with stored legumes for thousands of years, laboratory conditions are a good approximation of its 'natural' environment[79]. The experimental lines used for this study are described in detail in ref. 80 and ref. 39. Briefly, outbred stocks were initially collected from three geographic populations (Brazil, California and Yemen) and kept in laboratory conditions for >60 generations. The mitochondrial haplotypes of each population were introgressed in an orthogonal fashion into each nuclear background through repeated backcrossings over 16 generations, such that each mitochondrial haplotype existed in equal frequency on each nuclear background[81,82]. The introgressed lines were kept as separate populations for ca. 100 generations, at which point females from each line were backcrossed to males from the source population with the same nuclear background to refresh nuclear genetic diversity. These newly backcrossed lines were used as the three base populations (Brazil, California and Yemen) for our experimental evolution. At the time of study, these stocks had been reared on black-eyed beans (*Vigna unguiculata*) under standard laboratory rearing conditions of 29 °C in 12:12 h light/dark cycles and 50% relative humidity, and $N$ = 300–400, for >200 generations.

### Ancestral thermal performance

To assess the temperature tolerance and preference of the three ancestors, we measured how laboratory fitness changed with rearing temperature. Laboratory fitness was defined as the number of adult offspring produced per mating couple, divided by the egg-to-adult development time of those offspring as an estimate of generation time; this measure should thus closely correspond to each population's predicted intrinsic population growth rate. Each population was allowed to lay $F_1$ eggs on the standard host, black-eyed beans (*V. unguiculata*), at 29 °C. The beans were then split among 23, 29 and 35 °C. Once $F_1$ adults emerged, 20–30 mating couples per temperature were isolated and allowed to mate and lay eggs for their entire lifespan. Once emerged, $F_2$ adult offspring were recorded for their development time, frozen and later counted. To also get estimates of laboratory fitness at more extreme temperatures, ten $F_1$ mating couples emerging from 35 and 23 °C were moved to 37 and 17 °C, respectively, and assayed. We note here that, because these $F_1$ parents were not developing at the extreme temperatures, effects on their fertility from developing at the temperature extremes are not accounted for. This may have underestimated the stress imposed by 37 °C presented in Fig. 1. Thermal performance curves presented in Fig. 1 were fitted using a third-degree polynomial using the 'spline' function in R. When doing so, we fixed laboratory fitness at 38 °C to zero, as we have not been able to propagate any populations of *C. maculatus* at this temperature[38,83].

### Experimental evolution

Each replicate line was maintained on 250 ml black-eyed beans in a 1 l glass jar kept in a climate cabinet set at 50% relative humidity at the given evolution regime temperature (23 or 35 °C; Fig. 1a). Once adults emerged from beans, 600 beetles were transferred to a new jar with fresh beans and allowed to mate and then lay eggs until their death.

Beetles were transferred at the peak of hatching, so as not to apply strong direct selection for faster development time. See ref. 80 and ref. 39 for further details.

### Phenotypic data and analysis

We re-analysed previously collected data on thermal adaptation in seven life-history traits of females[39], including three core traits: LRS, adult weight and juvenile development time (Fig. 2a); as well as four rate-dependent traits measured over the first 24 hours of female reproductive lifespan: weight loss, water loss, early fecundity (eggs laid over the first day of adult female reproduction) and mass-specific metabolic rate (ml $CO_2$ $min^{-1}$ $mg^{-1}$). As reported by ref. 39, hot-adapted lines have evolved greater body mass, weight loss and reproductive output than cold lines, whereas rate-dependent traits show signs of local adaptation to temperature signified by statistically significant regime-by-assay temperature interactions.

These data were collected at generation 45 for cold-adapted lines and 60 for hot-adapted lines, but at the same time in a large common garden experiment removing (non-genetic) parental effects and including the two assay temperatures corresponding to the experimental evolution regimes (23 and 35 °C). Phenotypic data from the ancestral populations were not collected at the time of line establishment, but were scored in a separate experiment. Because each line had been lab-adapted for >200 generations, we assume that minimal lab adaptation had occurred after thermal regime establishments, and that these phenotypic data approximate the phenotypes of the founding populations. To estimate any block effects due to differing sampling times, a reference line was used in both the experiment on ancestors and evolve-and-resequence lines, and the relative difference in trait means of this reference line in the two experiments was used to standardize the ancestral data to minimize the possibility that block effects were wrongly assigned as evolved differences between ancestors and evolved lines. We note that the standardization was made for each trait by averaging over temperatures, and thus does not contribute to the differences in the repeatability and magnitude of phenotypic adaptation between the temperatures and genetic backgrounds that we study here. Data on LRS were then further gathered in a large second common garden experiment including both ancestors and evolved lines, after 80 (cold), 120 (hot) and 125 (ancestors) generations of evolution. Methods for data collection are thoroughly described in ref. 39.

By including ancestral data, we were able to assess the direction and magnitude of phenotypic evolution. Following ref. 84, we define parallel evolution here in a geometric sense by the angles of phenotypic change vectors between evolving populations. Although it should be emphasized that (non)parallelism is a continuous measure, three general categories can be made of the degree of (non)parallelism. In multi-dimensional trait space, evolution is parallel when the vectors of phenotypic change between evolving populations result in small angles ($\theta < 90°$) and anti-parallel when angles are large ($\theta > 90°$). Evolution can also result in orthogonal (that is, uncorrelated) changes when $\theta \approx 90°$. Following ref. 74, we calculated the $6 \times 6$ inter-population correlation matrix of evolutionary change vectors for each regime, and subsequently arccosine transformed correlations to angles. Because change vectors for each of the two replicate lines of a given origin were calculated using the phenotypic scores of the same ancestral population, shared measurement error in the ancestor will have contributed to more similarity between line replicates of the same origin, relative to similarity in phenotypic change vectors of lines of different origin (with separate ancestors used in calculations of change vectors). The angle between two change vectors can be decomposed into its components: $\theta = \cos^{-1} \frac{\mathbf{a}_x \mathbf{b}_x + \mathbf{a}_e \mathbf{b}_e}{\sqrt{(\mathbf{a}_x^2 + \mathbf{a}_e^2)(\mathbf{b}_x^2 + \mathbf{b}_e^2)}}$, where $\mathbf{a}_x$ and $\mathbf{b}_x$ represent the vectors of population mean trait change for evolved lines $a$ and $b$, and $\mathbf{a}_e$ and $\mathbf{b}_e$ represent the vectors of standard errors for each trait in the ancestors. This latter error component of the dot product (numerator) sums

to zero when measurement errors are uncorrelated, such as the case when using different ancestors in calculations, but equals the squared standard error when using the same ancestor. To produce unbiased comparisons within and between backgrounds we therefore corrected for this contribution of shared measurement error by subtracting this error component from the within estimates of change vectors. For the two cold-adapted Brazil populations, this error component was larger than the observed mean trait change. The angle of phenotypic change between these populations was therefore not estimated due to the statistical uncertainty.

By comparing the phenotypic distances between populations at the start ($S_d$) of experimental evolution (that is, between ancestors) to the distance at the end ($E_d$; that is, between the evolved lines), we estimated whether populations were diverging or converging phenotypically over the course of the experiment. These pairwise distances were calculated based on all seven phenotypic traits. When evolved lines are more distant than the ancestors in trait space ($E_d - S_d > 0$), populations are diverging, while the opposite ($E_d - S_d < 0$) is indicative of convergence. To estimate evolutionary responses to heat, we compared the hot lines to their ancestors when reared at 35 °C. Similarly, cold adaptation was estimated by comparing cold lines to their ancestors when reared at 23 °C. To perform the evolutionary change vector analysis, we first mean-scaled all traits into unitless measurements for comparison with each other by dividing ancestral and evolved estimates by the grand mean across all ancestors measured at the same assay temperature. Hence, rates of evolution are reported as proportional changes relative to ancestral trait means (see refs. 85,86). We then calculated the vector of phenotypic change from the ancestral to evolved samples in the seven-dimensional trait space. Similarly to the error correction conducted for angles, divergence within backgrounds will be overestimated relative to that between backgrounds due to the use of a single shared ancestral population in the former. We therefore corrected within-background distances by subtracting the squared standard errors of distance estimates in the two evolved replicates as $E_d = \sqrt{(\mathbf{a}_x - \mathbf{b}_x)^2 - (\mathbf{a}_e^2 + \mathbf{b}_e^2)}$, where $\mathbf{a}_x$ and $\mathbf{b}_x$ represent vectors of trait means, and $\mathbf{a}_e$ and $\mathbf{b}_e$ represent the vectors of standard errors, in evolved lines. For instances in which the error component was larger than the observed mean trait change component, the distances were set to zero (in all within-origin comparisons at 23 °C and between the two replicate California lines at 35 °C).

### Genomic data and analysis

For evolved populations, we isolated genomic DNA from pools of 60 male adults between generations 57–59 for cold lines and generations 67–69 for hot lines. Within each thermal regime, lines were sampled at the same time but varied in the number of generations of adaptation due to differences in development rate (Fig. 1a). Owing to a paucity of beetles remaining after line establishment, we were not able to sample the direct establishing ancestor. Instead, the third generation of heat-adapting lines was sampled and sequenced as a proxy of the ancestors (here simply referred to as 'ancestors' when considering genomic data); hence, comparisons between evolved and ancestral lines represent 60–62 and 64–66 generations of divergence for the cold and hot regime, respectively. Adult beetles were collected and stored at −80 °C. We purified DNA from the beetles using the KingFisher Cell and Tissue DNA Kit (Thermo Fisher Scientific). DNA fragment libraries for whole-genome sequencing were generated at the Science for Life Laboratory (Stockholm, Sweden) from each pool by following the standard Illumina TrueSeq PCR-free library prep protocol in which 350 base pair (bp) fragments were retained. Purified genomic libraries were then sequenced on the Illumina NovaSeq 6000 platform. In total, two lanes of 150 bp paired-end reads were generated.

We used BWA mem v.0.7.9 (ref. 87) to align the ~5.8 billion ~150-bp DNA sequences to a genome assembly for *C. maculatus* (accession: PRJEB60338). After sorting, merging and de-duplicating (Picard Tools

v.2.23.4, Broad Institute) our alignments, we then identified SNPs using the Genome Analysis Toolkit (GATK) best practice recommendations[88]. Variants were initially filtered based on GATK hard-filtering recommendations after manual and visual inspection of the data with the following parameters: QUAL < 20, QD < 2, FS > 20, SOR > 5, MQ < 40, −5 > MQRankSum > 5, −5 > ReadPosRankSum > 5. A second round of filtering was performed on regions of the genome that do not uniquely map to itself using GenMap v.2.4.1 (with parameters K31, E2)[89], sites with missing data across any sample, or DP < 20 across any sample. We retained 10,611,665 variants after filtering (~10 variants per kilobase).

To measure the strength of directional selection, we first identified candidate loci consisting of putatively selected SNPs. To determine whether a SNP is probably under selection, we incorporated temporal information of allele frequency change from the ancestor to evolved hot and cold lines. We estimated the variance effective population size ($N_e$) during the experiment for every replicate from patterns of allele frequency change from the total set of ~10 million SNPs using the R package poolSeq[90]. Given the estimated values of $N_e$, we then parameterized and tested a null model of evolution by genetic drift to determine the probability of observing the magnitude of allele frequency change for each SNP in each population. We calculated the probability of the observed allele frequency change from the ancestral to evolved populations based on a beta approximation to the basic Wright–Fisher model[91]. Following ref. [92], we assumed $p_t | p_0 \approx \mathrm{beta}(\alpha + 0.001, \beta + 0.001)$, where $\alpha = p_0 \frac{1-F}{F}, \beta = (1 - p_0)\frac{1-F}{F}$, $p_0$ and $p_t$ are the allele frequencies at the beginning and end of the interval, $F = 1 - (1 - \frac{1}{2N_e})^t$ and $t$ is the number of generations between samples. We retained SNPs with allele frequency changes more extreme than the 0.01th or 99.99th percentiles of the null distribution per SNP in any population (Fig. 2b), resulting in a total set of 475,194 putatively selected sites among all replicates. Estimation of selection coefficients was performed with Clear[93]. We generated the input for Clear from the set of putative SNPs with scripts from the PoPoolation2 pipeline[94]. Estimations of $N_e$, null model drift estimates and selection coefficients all require information on the number of generations elapsed between time points; for each replicate, we therefore added or subtracted three generations for cold- and hot-adapted lines, respectively, to the generation parameter to account for drift that occurred between ancestral lines and our proxies of the ancestors.

Repeatability at the SNP level was measured, similarly to the phenotypic data, by geometric estimates of $\theta$ and divergence ($E_d - S_d$) of allele frequency change vectors between populations of the set of putatively selected sites. Note that because two replicate ancestral samples were available per background, error corrections were not implemented. Because we are interested in parallelism due to selection, we focused on the repeatability of evolutionary change occurring in the set of putatively selected SNPs.

We additionally scanned for genomic regions that may exhibit parallel or anti-parallel responses between replicate lines per regime with the software AF-vapeR (v.0.2.1)[95] implemented in R. We followed standard methods, calculating allele frequency vectors in windows of 200 SNPs each. Null expectations of allele frequency change were conducted by permuting allele frequency changes among populations. A total of 10,000 permutations were conducted across all chromosomes, with the number of permutations per chromosome proportional to the size of the chromosome.

**Repeatability and historical contingency for different classes of alleles.** To explore if certain types of variants are more likely to contribute to repeated instances of thermal adaptation, we first identified SNPs that were assigned as being under selection in both replicate lines deriving from a particular background and regime. Because of this stricter criterion, we relaxed our null drift expectations to include SNPs with allele frequency changes more extreme than the 1st or 99th percentiles of the null distribution, resulting in 271,515–617,395 candidate SNPs

being selected. For each genetic background and evolution regime separately, SNPs were then assigned into the four categories based on the patterns of evolutionary change observed between regimes (synergistically pleiotropic, antagonistically pleiotropic, private cold and private hot; Fig. 4a). We then compared the relative abundance of SNPs that showed overlap between the three genetic backgrounds for each category. The same analysis was carried out at the gene level based on if the SNP fell within the protein-coding region of a gene. The biological processes for each gene set were analysed for enrichment using the R package topGO (v.2.38.1) and weight01 algorithm.

For each SNP category, we tested whether there was more overlap between backgrounds than expected by chance by performing forward in time drift simulations from all ancestral populations (R package poolSeq v.0.3.5.9). During each simulation iteration (n = 1,000), 10,000 random SNPs were sampled to parameterize all lines. Each line was parameterized with the ancestral allele frequencies, the number of generations evolved and the effective population sizes of respective lines.

**Genomic and phenotypic offsets.** To assess our ability to predict phenotypic evolution from genomic data, we calculated 'offsets' for both kinds of data, describing the level of genetic and phenotypic maladaptation relative to a well-adapted reference population[52,53]. Offsets were calculated for each geographic origin and evolution regime separately. Thus, for each background–regime combination, the reference was chosen as the line replicate with the highest laboratory fitness at the temperature to which it had been adapting. As for our estimates of ancestral thermal performance (Fig. 1), we here defined our proxy of laboratory fitness as the ratio between lifetime adult offspring production and their egg-to-adult development time, roughly corresponding to the line's intrinsic population growth rate. Thus, for example, for hot-adapted lines, Yemen 8 had higher fitness (3.65) than Yemen 7 (2.88) at 35 °C and was chosen as the reference line for all six Yemen populations phenotyped at 35 °C.

For our fitness offset, we used our metric of laboratory fitness, $\frac{\mathrm{LRS}}{\mathrm{Dev.\ time}}$, to calculate relative fitness of each tested line relative to its reference line. Phenotypic divergence offsets were calculated as the Euclidean distance in the seven mean-scaled traits between each tested line and its reference. The set of SNPs used to calculate genomic offsets were those putatively selected for in the reference line (more extreme than the 0.01th or 99.99th quantiles of the null drift distributions; see above). Genomic offsets were calculated as the sum of absolute differences between the reference and tested lines. The difference in each SNP's allele frequency was scaled by the absolute selection coefficient of that SNP calculated in the reference population such that SNPs with large differences in allele frequency but relatively small selection coefficients would not contribute greatly to the genomic offset.

**Reporting summary**

Further information on research design is available in the Nature Portfolio Reporting Summary linked to this article.

## Data availability

Phenotypic data are available via Dryad at https://doi.org/10.5061/dryad.bzkh189kd (ref. 96). The raw read data are available at the European Nucleotide Archive with the accession code PRJEB86644. All scripts and data fall under a CC0 1.0 Universal (CC0 1.0) licence.

## Code availability

Code for analyses are available via Dryad at https://doi.org/10.5061/dryad.bzkh189kd (ref. 96).

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

## Acknowledgements

This work was funded by grants 2019-05024 from the Swedish Research Council (VR) and 2022-01117 from Formas to D.B., and Swedish Research Council grants 2017-04963 and 2022-03427 to R.S. The storage and computations for this work were enabled by resources in projects SNIC 2021/5-125 and SNIC 2020/6-175 provided by the Swedish National Infrastructure for Computing (SNIC) at UPPMAX, partially funded by the Swedish Research Council through grant agreement no. 2018-05973. Special thanks to K. Tunström, R. Steward and K. Roberts for critiques/suggestions, and L. von Schmalensee and M. Metz for contributing data for thermal performance curves.

## Author contributions

D.B., R.S. and A.R. conceived the study. C.G.-T., J.B. and M.d.l.P.C.-M. collected the data. A.R. and D.B. analysed and wrote the paper with input from all co-authors.

## Funding

## Competing interests

The authors declare no competing interests.

## Additional information

**Extended data** is available for this paper at https://doi.org/10.1038/s41559-025-02716-5.

**Correspondence and requests for materials** should be addressed to Alexandre Rêgo.

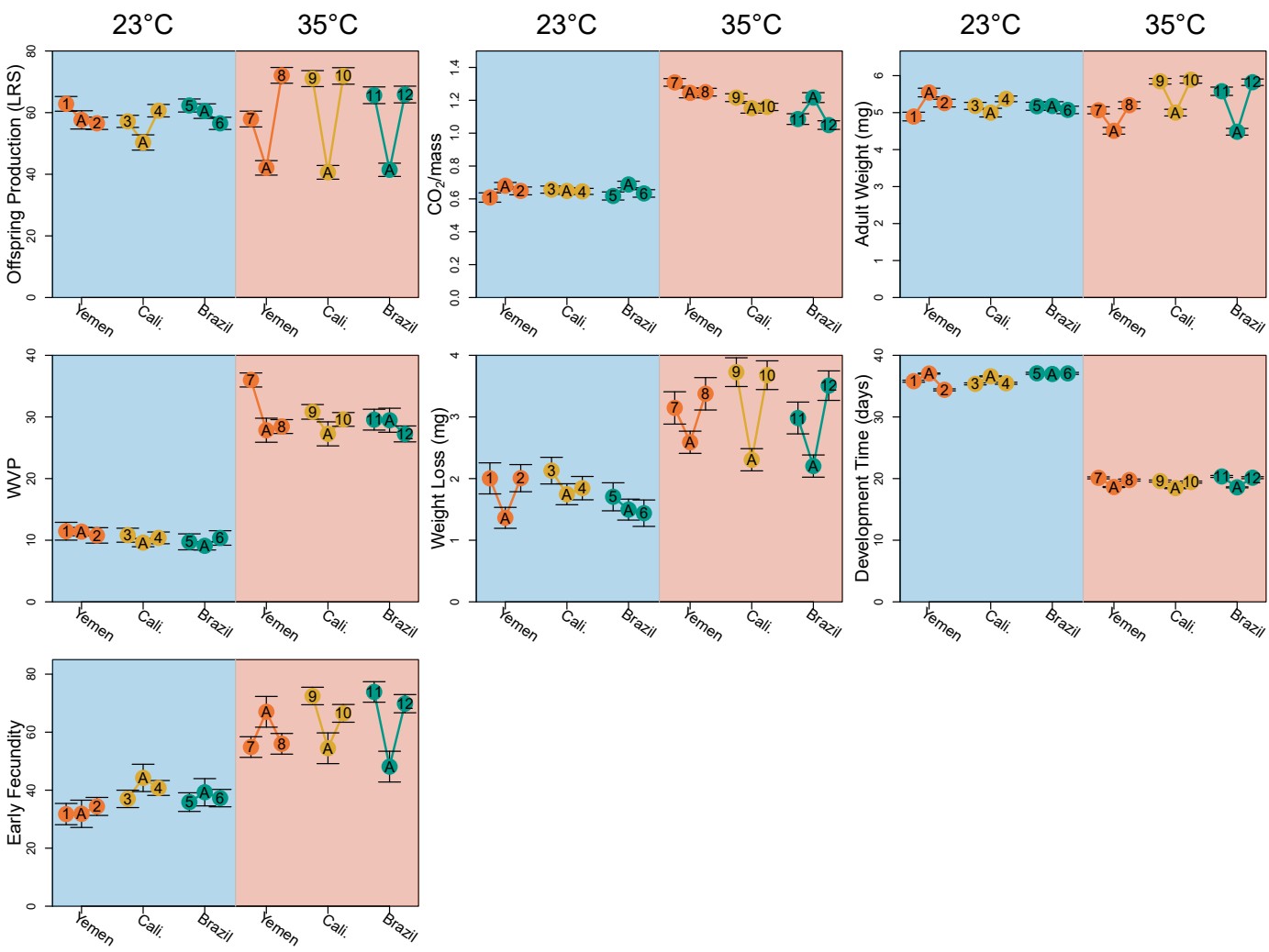

**Extended Data Fig. 1 | Phenotypic evolution of 7 life-history traits.** Means and one standard error for the seven life-history traits in evolved and ancestral lines. Note that ancestors ('A') and evolved lines (1-12) were measured in their evolved temperature regime, denoted by background color.

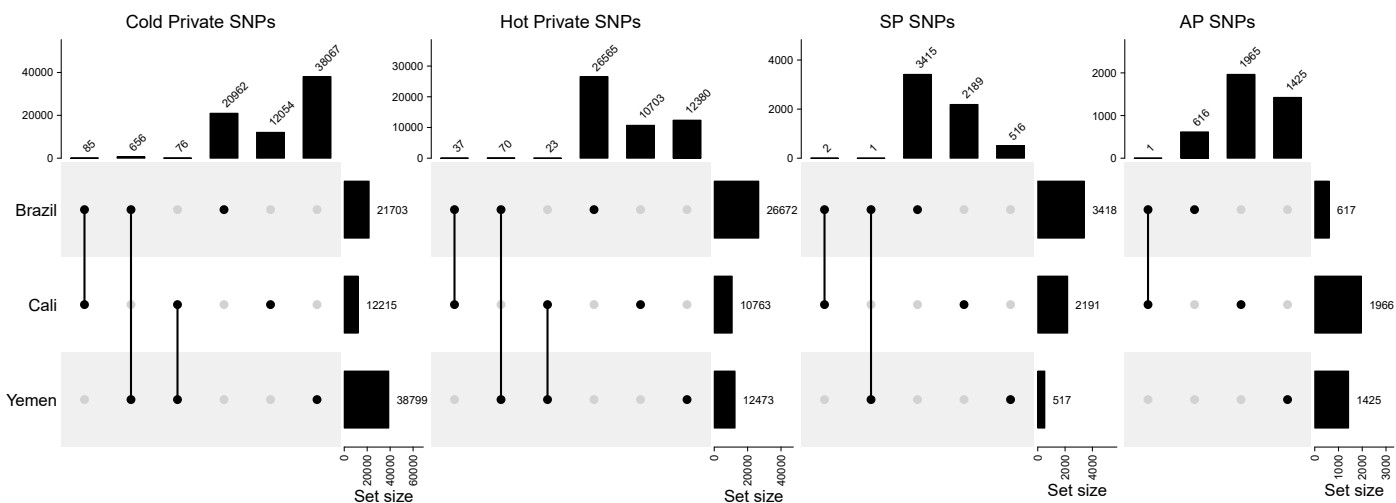

**Extended Data Fig. 2 | Upset plots of different categories of allele frequency change among thermal regimes.** Cold Private (selected only at cold), Hot Private (selected only at hot), AP (antagonistically pleiotropic - selected in opposite directions in the hot and cold regime) and SP (Synergistically pleiotropic - selected in the same direction in the hot and cold regime).

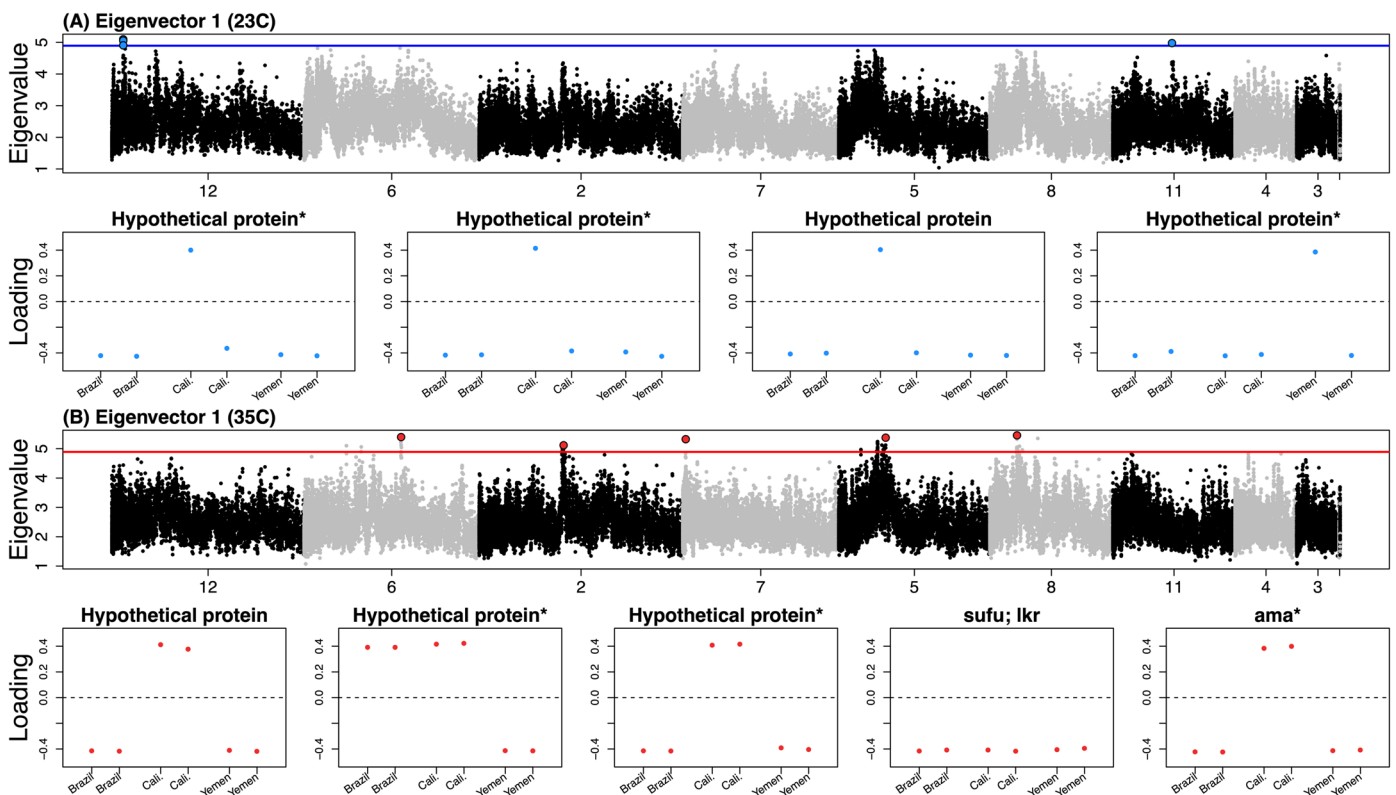

**Extended Data Fig. 3 | Eigen analyses of genomic regions showing parallel regions of allele frequency change.** Manhattan plots showing the Eigenvalue for regions (window size = 200 SNPs) of the genome across all 6 cold-adapted (**A**) or hot-adapted (**B**) lines. Windows exceeding the 99% significance threshold are highlighted- all 4 independent chromosomal windows for 23 °C (below panel **A**) and the top 5 significant windows for 35 °C (below panel **B**). For each significant window (presented in left-to-right order), we show population loadings on Eigenvector 1, with genes in or nearest to each window labeled (nearest genes marked with asterisks).

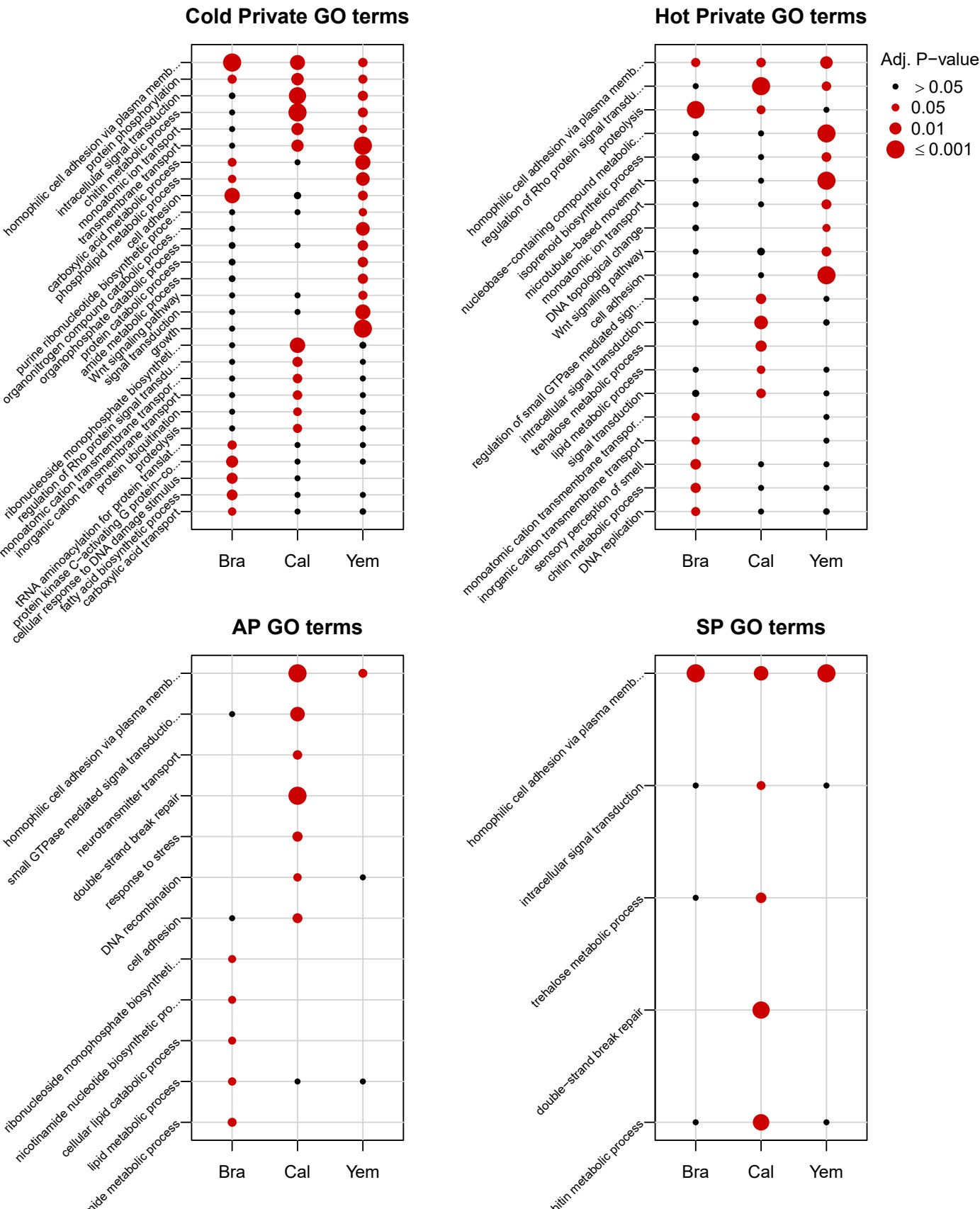

**Extended Data Fig. 4 | Dot plot indicating significant GO terms per gene set category (based on Fig. 4).** Adjusted p-values equal to 1 are not shown.

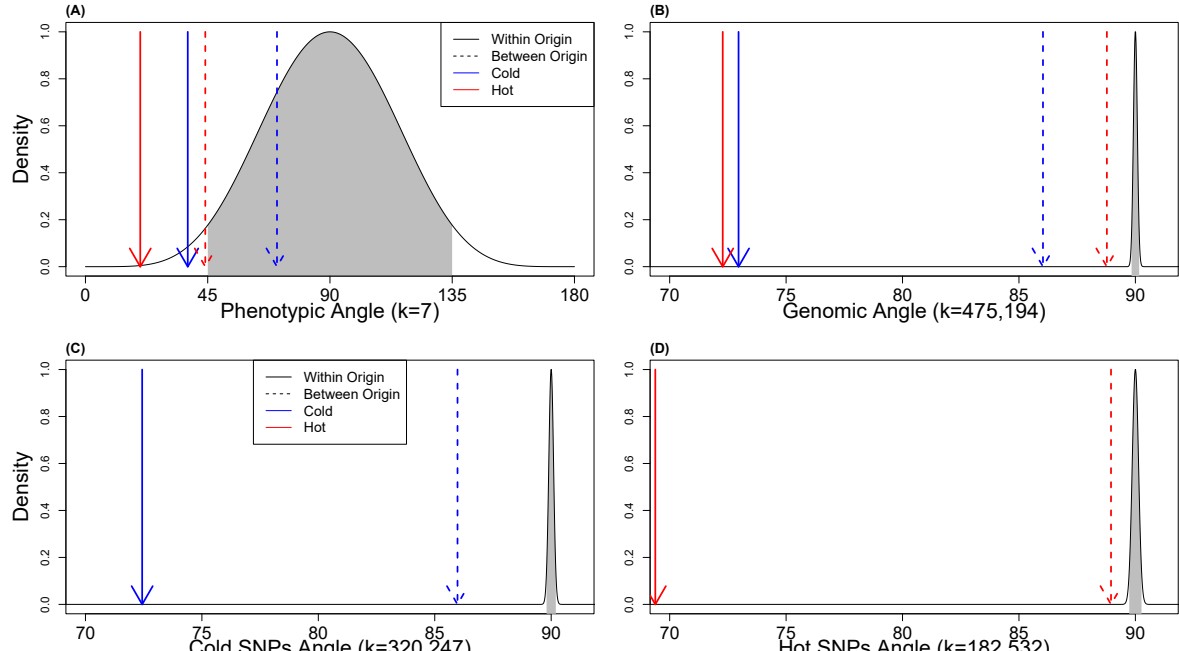

**Extended Data Fig. 5 | Null distributions of random angles in k-dimensional space.** Distributions shown are associated with (**A**) phenotypic parallelism, (**B**) genomic parallelism using any selected site, and (**C & D**) genomic parallelism using sites selected for per regime. The 95% confidence intervals associated with random angles are shown shaded in grey. The mean within- and between-origin comparisons are presented as arrows and colored per temperature regime.

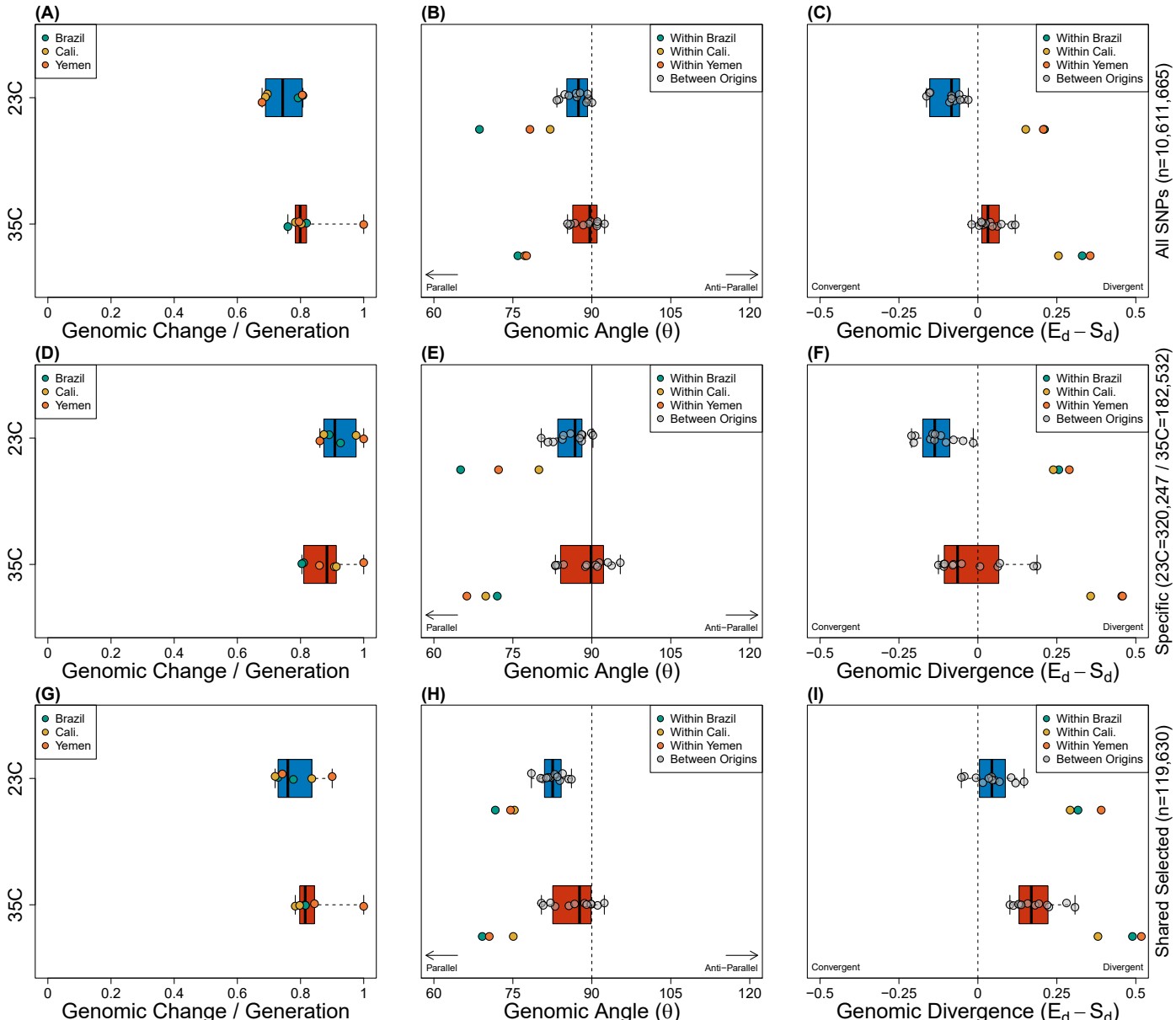

**Extended Data Fig. 6 | Evolutionary rates, pairwise angles, and measurements of divergent evolution between different selection criteria of SNPs.** Allele frequency changes found in all SNPs called (top row, "all"), SNPs selected for in each temperature regime independently (middle row, "specific"), and SNPs which are both selected for in any line but also polymorphic among all ancestors (bottom row, "shared"). Evolutionary rates (**A** & **D** & **G**) are given for individual lines, while angles (**B** & **E** & **H**) and divergences (**C** & **F** & **I**) are given for pairwise comparisons between populations of different (grey points) and same (colored points) genetic background. Evolutionary rates have been scaled by the maximum evolutionary rate in the dataset, while divergences have been scaled by the distance between the two most differentiated ancestors.

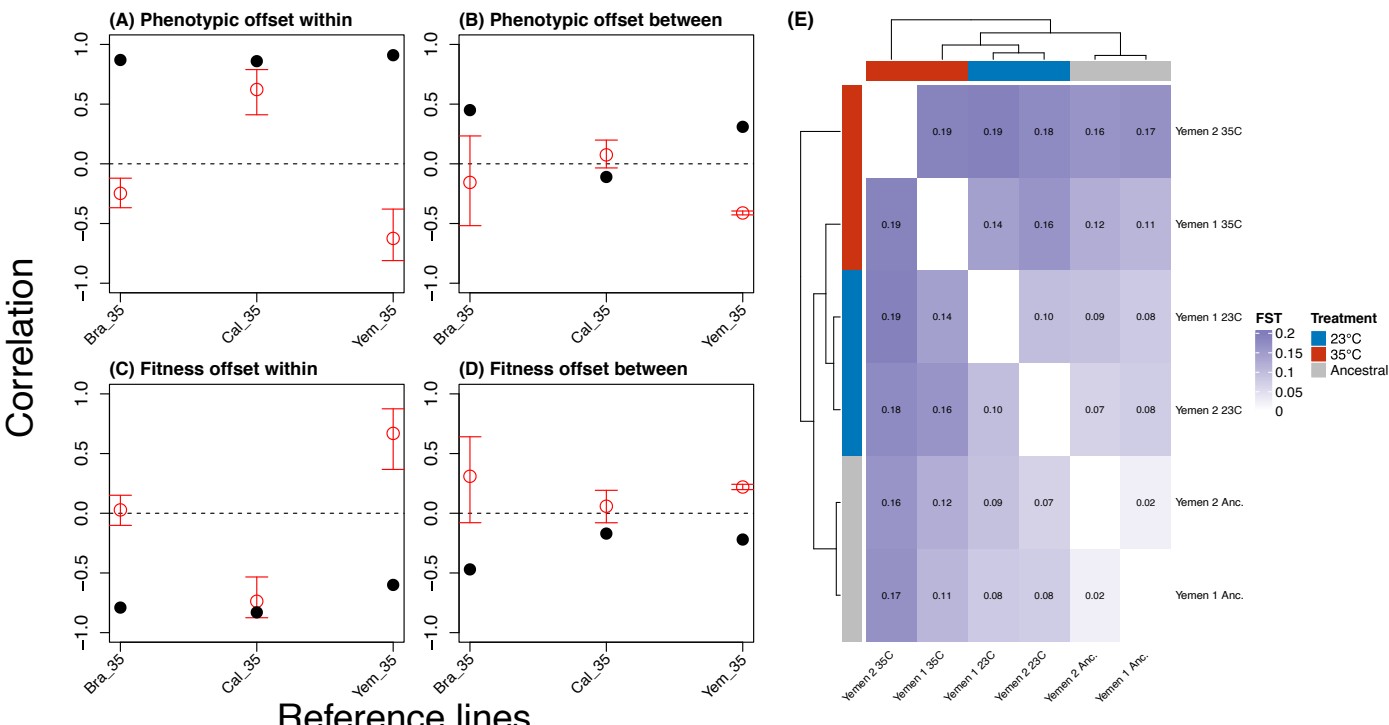

**Extended Data Fig. 7 | Bootstrapped correlations for genomic offsets at hot temperature using sets of randomly chosen SNPs.** Genomic offsets (**A**-**D**) were calculated 1,000 times using randomly sampled SNPs of equal number to the SNPs used in hot line offsets (n=10,546-10,649). The means and 0.025th and 0.975th quantiles for bootstrapped correlations are shown as colored points and lines, respectively. The correlation based on selected SNPs is shown as a solid black point (see Fig. 6). Correlations are separated by within (**A**, **C**) and between (**B**, **D**) origins. Predictions based on candidate SNPs outperform those based on randomly selected SNPs. Note that predictions based on random SNPs sometimes go in opposite direction to those based on selected SNPs (for example, see predictions within the Yemen background), yielding highly

inaccurate predictions. This may happen when several isolated populations adapt to a particular environment (for example, heat) and experience stronger selection than populations evolving in another environment (for example, cold). When the case, the populations under stronger selection can show high phenotypic similarity but diverge more from each other at neutral sites due to stronger genetic drift and draft during the processes of adaptation. As a result, genomic distances (and offsets) based on neutral variation may be larger between populations adapting in parallel (illustrated by whole-genome FST for the Yemen background; **E**). Thus, sometimes the use of neutral or genome-wide variation may result in counterintuitive, and even misleading, predictions of local adaptation.

# Reporting Summary

## Statistics

For all statistical analyses, confirm that the following items are present in the figure legend, table legend, main text, or Methods section.

| n/a | Confirmed | |
|---|---|---|
| ☐ | ☒ | The exact sample size (*n*) for each experimental group/condition, given as a discrete number and unit of measurement |
| ☐ | ☒ | A statement on whether measurements were taken from distinct samples or whether the same sample was measured repeatedly |
| ☐ | ☒ | The statistical test(s) used AND whether they are one- or two-sided *Only common tests should be described solely by name; describe more complex techniques in the Methods section.* |
| ☐ | ☒ | A description of all covariates tested |
| ☐ | ☒ | A description of any assumptions or corrections, such as tests of normality and adjustment for multiple comparisons |
| ☐ | ☒ | A full description of the statistical parameters including central tendency (e.g. means) or other basic estimates (e.g. regression coefficient) AND variation (e.g. standard deviation) or associated estimates of uncertainty (e.g. confidence intervals) |
| ☐ | ☒ | For null hypothesis testing, the test statistic (e.g. *F*, *t*, *r*) with confidence intervals, effect sizes, degrees of freedom and *P* value noted *Give P values as exact values whenever suitable.* |
| ☒ | ☐ | For Bayesian analysis, information on the choice of priors and Markov chain Monte Carlo settings |
| ☒ | ☐ | For hierarchical and complex designs, identification of the appropriate level for tests and full reporting of outcomes |
| ☐ | ☒ | Estimates of effect sizes (e.g. Cohen's *d*, Pearson's *r*), indicating how they were calculated |

*Our web collection on statistics for biologists contains articles on many of the points above.*

## Software and code

Policy information about availability of computer code

| Data collection | R and Rstudio (v. 2022.07.2 Build 576). Open source. See "Code availability" statement in manuscript. |
|---|---|
| Data analysis | R and Rstudio (v. 2022.07.2 Build 576). Open source. Code available at Dryad: https://doi.org/10.5061/dryad.bzkh189kd Peer-reviewer link to data and code: http://datadryad.org/stash/share/ioNUncvnoFT2hGKqcIYFKKIunBITTfC75W0HMzjmFCo |

For manuscripts utilizing custom algorithms or software that are central to the research but not yet described in published literature, software must be made available to editors and reviewers. We strongly encourage code deposition in a community repository (e.g. GitHub). See the Nature Portfolio guidelines for submitting code & software for further information.

## Data

Policy information about availability of data

All manuscripts must include a data availability statement. This statement should provide the following information, where applicable:
- Accession codes, unique identifiers, or web links for publicly available datasets
- A description of any restrictions on data availability
- For clinical datasets or third party data, please ensure that the statement adheres to our policy

European Nucleotide Archive acession code: PRJEB86644

# Research involving human participants, their data, or biological material

Policy information about studies with [human participants or human data](). See also policy information about [sex, gender (identity/presentation), and sexual orientation]() and [race, ethnicity and racism]().

| Reporting on sex and gender | N/A |
|---|---|
| Reporting on race, ethnicity, or other socially relevant groupings | N/A |
| Population characteristics | N/A |
| Recruitment | N/A |
| Ethics oversight | N/A |

Note that full information on the approval of the study protocol must also be provided in the manuscript.

# Field-specific reporting

Please select the one below that is the best fit for your research. If you are not sure, read the appropriate sections before making your selection.

☐ Life sciences  ☐ Behavioural & social sciences  ☒ Ecological, evolutionary & environmental sciences

For a reference copy of the document with all sections, see [nature.com/documents/nr-reporting-summary-flat.pdf]()

# Ecological, evolutionary & environmental sciences study design

All studies must disclose on these points even when the disclosure is negative.

| Study description | 12 Experimental evolution lines and their 3 ancestal lines, of the seed beetle Callosobruchus maculatus, were analyzed. 6 lines were evovled at hot (35C) temperature, and 6 lines were evolved at cold (23C) temperature. The ancestors were kept at ancestral lab temperautre (29C). The evolved lines were created from the ancestors, that originally were sampled from Brazil, Yemen and California (USA). Thus experimental evolution was carried out on three different genetic backgrounds, with two biological replicates per evolution regime and and background.

Evolution lines were scored for their life history adaptation by measuring seven traits at generations 45-60 and again for lifetime offspring production at generations 80-120. Ancestors were measured for their traits at generation 120. Genomic sampling was also conducted at generation 60 in evolved lines and at generation 0 in the ancestor.

We quantified repeatability in both phenotypic and genomic evolution by calculating using vector analyses of evolutionary change. We subsequently modeled the predictability of fitness and phenotypic traits from genomic data across and between geographic origins. |
|---|---|
| Research sample | Callosobruchus macualtus seed beetles (as desribed above). 12 evolved + 3 ancestral lines. Most measurements were done on females, as these are more directly affecting population growth rates. |
| Sampling strategy | We measured several thousands of beetles to attain relatively accurate estimates of means for all 7 life-history traits of each line. For these traits, our lab has measured them before, we therefore had a good idea of what was needed, even though no direct power analysis was conducted. Having that said, we also were limited practically, so we also pushed on and tried to measure life history traits for as many beetles as possible for the assays of metabolic rate and related traits.

We maximized statistical power of genomic sampling by pooling individuals from a particular line into DNA libraries. We sequenced pools of individuals to sufficient depth to have high accuracy in estimating allele frequencies (>20X coverage). |
| Data collection | Life-history traits
We quantified thermal adaptation in female life-history by measuring three core traits: lifetime reproductive success (LRS), juvenile development time and adult body mass, and four rate-dependent traits: early fecundity, weight loss, water loss, and mass-specific metabolic rate (ml $CO_2$/mass/min) over the first 16h of female reproduction. All life-history traits were collected at generations 40 for cold-adapted lines and 60 for hot-adapted lines, in a large common garden experiment including the two assay temperatures corresponding to the experimental evolution treatments (23°C and 35°C). Ancestral lines were scored in the same experimental conditions with the addition of the ancestral 29°C assay temperature, but on a later occasion following ca. 125 generations of experimental evolution. Note that the ancestors had been kept at the ancestral conditions, to which they had already adapted for more than 300 generations prior to the start of experimental evolution. It can therefore be assumed that the measured trait values correspond well with the trait values at the start of experimental evolution. To control for potential differences in the separate experiments on ancestors and evolved lines stemming from unknown sources, we reared an independent laboratory adapted reference population in both experiments. This indicated that differences in rearing had affected the life-history traits scored over the first 16h of reproduction. We therefore standardized the traits scored during respirometry of the three founding ancestors by this estimated amount (adult mass: increased by 6.4%, metabolic rate: reduced by 12%; early fecundity: reduced by 18%; water loss: |

reduced by 15%, and weight loss: reduced by 25%) in order not to erroneously assign these differences to evolutionary divergence between ancestors and evolved lines. Note that this was done averaged across the three assay temperatures and geographic origins. Therefore, our approach to provide more accurate measures of evolutionary divergence between evolved lines and ancestors did not affect the estimated temperature-dependence of adaptation or the importance of geographic differences.

Before assays of life-history traits, non-genetic parental effects were removed by moving F0 grandparents of the assayed individuals into a common temperature of 29°C to lay eggs. The emerging beetles in the next (parental) F1 generation were allowed to mate and lay eggs on beans provided ad libitum. Following 48 hours of egg laying, the beans were split and assigned to one of the two (for ancestors, three) assay temperatures. The emerging adult F2 offspring were phenotyped for their life-history (Fig. 4). Newly emerged (0-48 hours old) virgin females were mated to males by placing three males and females together in a petri dish over night at the assay temperature. In the following morning, the three females were weighed for their body mass and then placed together inside a glass vial filled with black eyed beans to be measured for their metabolic rate, water loss and early fecundity at their designated assay temperature. The glass vials were placed in a Sable Systems (Las Vegas, NV, USA) high-throughput respirometry system. Briefly, the respirometry was set up in stop-flow mode, and $CO_2$ production and water-loss was measured for up to 23 vials on a given experimental day. The first vial was left empty and served as a baseline to control for any drift of the gas analysers during each session. Vials were measured over 17 cycles, each of a length of 57.5 minutes. Mean metabolic rate and water loss for each vial was calculated across cycles 2-17, with the readings from the first cycle discarded (as it contains human-produced water and $CO_2$). After respirometry, females were weighed again to record their weight loss and beans with eggs were isolated and counted to record early fecundity. In total we followed 386 triads of females for the evolved lines and another 115 triads from their ancestors.

From the same rearing we measured egg-to-adult development time for two technical replicates per line and assay temperature, each consisting of 40-120 individuals. We calculated a mean development time per technical replicate and used this in analysis. We also collected virgin males and females and placed three males and three females together in a petri dish with ad libitum beans to record lifetime reproductive output (LRS) at each assay temperature. In total we recorded LRS for 258 couple triplets for evolved lines, and another 115 couple triplets for the ancestors. These data were complemented with additional data from both evolved and ancestral lines reared in a common garden design in two consecutive years (corresponding to generation 120/135 for ancestors, 115/130 for hot-adapted lines, and 80/90 for cold-adapted lines). In these rearings, a single male and female were put together in a petri dish with ad libitum host seeds, with otherwise the same conditions. For ancestors we scored 396 couples, and for evolved lines 789 couples, across both experimental years. LRS was analysed per female, hence we divided all offspring counts from female triads by three before analysis.

| | |
|---|---|
| Timing and spatial scale | Life history traits were scored in April-June 2018 (generations 45-60) for evolved lines, and for ancestors in April-May 2022. Because ancestors and evolved lines were not scored at the same time for these traits, we also included data for lifetime time offspring production measured in a common garden (including both) at generation January-March 2023.<br><br>DNA extraction and sequencing was also conducted during the same time that life history traits were measured (2018). |
| Data exclusions | N/A |
| Reproducibility | As described above, we measured lifetime offspring production twice to verify that differences remained between evolution regimes and ancestors, which resulted in concordant results between the two samples. |
| Randomization | We reared populations and always tried to run samples from each line on the same day. This was not possible to do on all days as the lines have evolved differences in development time, and beetles develop at different rates in different temperatures. Hence, while lines were started at the same time in the experiment, they were not finishing at the same time, and for some traits they were measured on different days. |
| Blinding | For all assays we used ID numbers and not the actual name of the lines. However, since beetles develop predictably from different temperature treatments, it was not possible to blind the observer from this particular aspect of the experimental design. |

Did the study involve field work? ☐ Yes ☒ No

# Reporting for specific materials, systems and methods

We require information from authors about some types of materials, experimental systems and methods used in many studies. Here, indicate whether each material, system or method listed is relevant to your study. If you are not sure if a list item applies to your research, read the appropriate section before selecting a response.

## Materials & experimental systems

| n/a | Involved in the study |
|---|---|
| ☒ | Antibodies |
| ☒ | Eukaryotic cell lines |
| ☒ | Palaeontology and archaeology |
| ☐ | ☒ Animals and other organisms |
| ☒ | Clinical data |
| ☒ | Dual use research of concern |
| ☒ | Plants |

## Methods

| n/a | Involved in the study |
|---|---|
| ☒ | ChIP-seq |
| ☒ | Flow cytometry |
| ☒ | MRI-based neuroimaging |

# Animals and other research organisms

Policy information about studies involving animals; ARRIVE guidelines recommended for reporting animal research, and Sex and Gender in Research

| | |
|---|---|
| Laboratory animals | Callosobruchus maculatus (strains from Brazil, Yemen, and USA). |
| Wild animals | N/A |
| Reporting on sex | Only females were measured. |
| Field-collected samples | Laboratory stocks which have been maintained >10 years before experiments (according to standard lab conditions - 29C and 50-55% RH) |
| Ethics oversight | No ethical approval is necessary for insects according to national legislation. |

Note that full information on the approval of the study protocol must also be provided in the manuscript.

# Plants

| | |
|---|---|
| Seed stocks | Report on the source of all seed stocks or other plant material used. If applicable, state the seed stock centre and catalogue number. If plant specimens were collected from the field, describe the collection location, date and sampling procedures. |
| Novel plant genotypes | Describe the methods by which all novel plant genotypes were produced. This includes those generated by transgenic approaches, gene editing, chemical/radiation-based mutagenesis and hybridization. For transgenic lines, describe the transformation method, the number of independent lines analyzed and the generation upon which experiments were performed. For gene-edited lines, describe the editor used, the endogenous sequence targeted for editing, the targeting guide RNA sequence (if applicable) and how the editor was applied. |
| Authentication | Describe any authentication procedures for each seed stock used or novel genotype generated. Describe any experiments used to assess the effect of a mutation and, where applicable, how potential secondary effects (e.g. second site T-DNA insertions, mosiacism, off-target gene editing) were examined. |

