## [Peer Review file · Nature Ecology & Evolution]

Temperature-specific repeatability of evolution and its implications for genomic predictions of adaptation to warming

Corresponding Author: Dr Alexandre Rego

Version 0:

Decision Letter:

12th November 2024

Dear Alexandre,

Your manuscript entitled "Strong Selection, but low repeatability: Temperature-specific effects on genomic predictions of adaptation" has now been seen by 3 reviewers, whose comments are attached. The reviewers have raised a number of concerns which will need to be addressed before we can offer publication in Nature Ecology & Evolution. We will therefore need to see your responses to the criticisms raised, along with a revised manuscript, before we can reach a final decision regarding publication.

We therefore invite you to revise your manuscript taking into account all reviewer comments. Please highlight all changes in the manuscript text file.

* If you have not done so already please begin to revise your manuscript so that it conforms to our Article format instructions at <http://www.nature.com/natecolevol/info/final-submission>. Refer also to any guidelines provided in this letter.

Link Redacted

Nature Ecology & Evolution is committed to improving transparency in authorship. As part of our efforts in this direction, we are now requesting that all authors identified as 'corresponding author' on published papers create and link their Open Researcher and Contributor Identifier (ORCID) with their account on the Manuscript Tracking System (MTS), prior to

acceptance. ORCID helps the scientific community achieve unambiguous attribution of all scholarly contributions. You can create and link your ORCID from the home page of the MTS by clicking on 'Modify my Springer Nature account'. For more information please visit www.springernature.com/orcid.

[redacted]

Reviewer expertise:

Reviewer #1: experimental evolution in microbes

Reviewer #2: correlated evolution, predictability of evolution, insects

Reviewer #3: insect ecophysiology, thermal tolerance

Reviewers' comments:

Reviewer #1 (Remarks to the Author):

Given the historical nature and role of random and stochastic processes in evolution, the predictability of evolutionary outcomes has been a topic of major interest in recent decades. The evidence largely suggests that contingency and predictability of outcomes varies depending on the organisms being examined and the conditions under which evolution takes place, and altogether indicates that much more study is needed to identify important factors involved. One suggestion from theory is that strong selection promotes predictability. This manuscript describes research designed to examine this suggestion. It specifically looks at the predictability of adaptation of seed beetles to elevated or depressed temperature during experimental evolution, with analysis done at the levels of genome sequence change to life history phenotypes. They hypothesized that elevated temperatures, as is seen with global warming, exerts a stronger selection that reduced temperatures, and that so they predicted seeing more predictable evolution in the elevated temperature conditions. They found that phenotypic evolution under elevated heat is more predictable, more parallel, and faster than under depressed ones, but that genotypic evolution was the reverse. These findings were contrary to what they had expected, and they conclude that they are owing to epistatic interactions that reduce the ability to predict phenotypic adaptation from genotypic data. Somewhat surprisingly given earlier work in microbial systems, they found that antagonistic pleiotropy was highly involved in thermal adaptation, which was interesting.

I found this manuscript to be clear, well-written, and compelling. The research it describes is very impressive, deep, and incisive. The findings are in line with previous work that has shown similar disparities in predictability of phenotypic adaptation versus lack of predictability in genotypic changes underlying that adaptation. The explanation they give that differences between the populations evolved under the different temperature regimes as owing to epistasis is compelling and well supported. It is in line with ongoing findings that epistasis is extremely important in adaptation, and plays strong roles in evolutionary trajectories during selection. Moreover, they found important differences in repeatability across genetic backgrounds, and, I think, rightfully conclude that this suggests a strong role for historical contingency. Indeed, this finding is in line with the emerging consensus that genetic relationships are very important to predictability of evolution across lineages. Unfortunately, as the authors note (and I agree with them), this means that it can be difficult to usefully generalize expected evolutionary patterns from limited numbers of study systems, which is problematic for developing models of how organisms will evolve in response to climate change.

To be honest, after going through the paper multiple times, I do not have anything to really call out or criticize. This is a superb piece of work. The mathematical and computational analyses are beyond my understanding, so I hope the other reviewer(s) can identify any defects. The experiments were well-conducted and clearly described, with issues such as variation in generations under the two conditions clearly noted and accounted for. The conceptual analysis is spot on, rigorous, and not overblown. It says a lot that, every time I had a question about some aspect of the work during my initial read through, I found it soon addressed later in the text. This paper shows a great deal of deep thought and careful consideration. Usually, I am able to at least point out bad grammar or other such that should be corrected, but, if such issues are present, I did not notice them. This is one of the rare instances in which I do not feel that I can make any suggestions that would help to improve the paper. This is an excellent manuscript that describes excellent and important work in this area, and I can see it becoming a well-cited part of the literature on evolutionary predictability and contingency.

Reviewer #2 (Remarks to the Author):

This paper combines phenotypic and genomic data from an experimental evolution design in beetles to assess how repeatable (or "parallel") evolutionary change is when adapting to hot and cold environments. They find that phenotypic change is more parallel when adapting to hot vs cold environments, but the genomic data indicates the opposite. Based on

this and some other results the authors suggest a potentially more important role for pleiotropy in adaptation to hot environments. This paper is interesting, and the dataset, analyses and set of questions addressed is impressive and certainly within the scope of a journal like NEE. However, I do have a few issues with how the analysis of repeatability was performed that I think need to be addressed before I will fully endorse the authors story here.

A fundamental analytical challenge with addressing these questions about repeatability of evolution is that it is very hard to interpret what an angle in high dimensional trait space means. Put simply, when we study a lot of traits at once and examine angles between evolutionary change vectors from replicate lineages evolving to a common gradient, we have the following problems: 1) it is extremely hard to intuitively interpret angles in any meaningful way (they are not an effect size), and if we examine many lineages and many traits, interpreting individual pairwise angles can fail to fully reveal clustering of vectors in certain directions of trait space, 2) it does not make sense to compare, even informally, two (or more) angles if the trait dimensionality between the two vector pairs differs, the angles simply are not comparable. The authors are clearly aware of these issues, but I don't think they have been adequately addressed.

For example, the main counterintuitive result of the paper is that angles for genomic data are more orthogonal in hot than cold, opposite the phenotype data. Yet, the authors found on average about twice as many SNPs under selection in Cold lines than in Hot lines (based on the supplemental table), but all of these SNPs were included in this analysis of angles. That is, in this analysis there were many more SNPs that showed no evidence of being under selection (or even more disturbingly, were even polymorphic in all of the ancestral populations!, see below) in hot vs cold. So we know that most of the SNPs showed no change in the hot lines, and when we include all of these SNPs sure enough we find that evolution was more repeatable in the cold lines. Put simply I think the result is purely an outcome of the fact that more SNPs were under selection in the cold lines, but all the SNPs were included in the analysis. One could make a philosophical argument that it would be valid to include all of the SNPs anyways, but given the authors are specifically filtering by those under selection and there is a huge discrepancy in the number under selection in each environment, this creates a problem for me with how we interpret these results.

The alternative to the above approach would be to filter SNPs separately by environment, and analyze parallelism in change separately for hot and cold, using the hot and cold SNPs. Of course then there would be different dimensionality and so the angles in the "hot" analysis could not be directly comparable to the "cold", but nonetheless there are several standardized effect sizes (Watanabe 2022) that could be presented to assess relative parallelism in adaptation to each environment. I think this would be preferable to the present approach, and could likely change the story entirely: if we are just focusing on the SNPs under selection in each environment, is evolution still more parallel in cold?

Second, I don't think the authors have been clear enough about how we really can't contrast the angles from the trait data with those from the genomic data. They have 7 phenotypic traits and over 400,000 genomic traits. The authors have done some simulations so are aware of the null expectations, I suggest they plot the nulls to see for themselves how even only the slightest departure from pure orthogonality carries great meaning when we have 400,000 traits; I doubt the Cis on the null expectation would even be visible when plotted on Figure 3E. Put simply the value of an angle from one of the trait vector correlations simply is not relevant when we are thinking about angles from the genomic comparisons. This point about dimensionality and interpretation of angles has been made a few times recently in the literature, but it can't be overemphasized. I actually think the genomic data might be more meaningfully "parallel" than the trait data, but it is hard to say just based on the analyses presented. All of these points should be made more obvious to readers and anyone looking at the figures.

Aside from the above issues, I think the assessment of genomic parallelism in particular is a bit simplistic. For example, Whiting et al 2022 (<https://doi.org/10.1111/2041-210X.13952>) extended De Lisle and Bolnick's eigen analysis approach to understand more complex patterns of parallel change in SNP data. Given there are 400000 SNPs here, I wouldn't be surprised if the patterns of clustering of these vectors are more complex than can be assessed by inspection of the angles.

Finally, I struggled a bit to understand the point of the "fitness offset" analysis. It seems to be simply trying to correlate distance in a fitness measure (which is actually more of a life history measure than fitness, see below) with SNP distance. There was little conceptual or theoretical motivation provided for this, which is discomfoting especially considering that the measure of fitness is not necessarily even close to what most evolutionary biologists think of as fitness.

Specific/minor comments

I suggest the authors try to reduce acronym use. We have PH, PC, SP, AP, AW, WL and more. Just write out the words, please, I had to think about what these meant each time I came across them

Line 178 A very minor point: Treating this type of data as a correlation matrix was actually proposed by De Lisle and Bolnick (Ref 98), Watanabe proposed some different ways to analyze the correlation matrix

Line 200-226 This description was a bit hard to follow. I think the authors are comparing pairwise distances in 7-dimensional trait space, but it could be a trait-by-trait analysis as the description is unclear.

Line 336 I have never seen such a measure of fitness, and this sounds more like some description of life history than of fitness itself, as it will measure variation in how resources are allocated to offspring production through time (i.e., life history). Lifetime reproductive success is what most people think of as fitness, and this can be multiplied by a juvenile survival estimate if that is available.

Line 370-379 This is also a nice demonstration of why it might often be useful to think of parallelism and convergence as separate but related phenomenon

Line 467 I think “partly” is a severe understatement; you literally have hundreds of thousands of SNPs, and just 7 phenotypic traits. See above comment

Line 504 Wait, why wasn't the main analysis based on this subset to begin with? It makes no sense to me to assess parallelism across a set of SNPs that weren't segregating in some of the ancestral populations, since unless there is significant mutational input (doubtful in a relatively short timespan in an experiment with small N_e) then we wouldn't expect to see any change at these loci anyways!!

Reviewer #2 (Remarks on code availability):

I downloaded the scripts and data, and didn't run the whole thing but did check some of the matrix calculations. It does appear all the data and code are presented, and it is pretty clear and easy to interpret their scripts. I think they have done a fine job on that

Reviewer #3 (Remarks to the Author):

In this manuscript, the authors examine the repeatability of evolution to high and low temperatures in a seed beetle. They find that phenotypic evolution was faster and more repeatable in the heat, but that the opposite was true for genotypic change, and that there was a strong effect of genetic background. I found the manuscript generally clear and easy to read, and I commend the authors for their thorough investigation.

1. One challenge I have with this manuscript is the relatively low population size. Insect population sizes can be in the millions, and yet here the population size is 215 (and indeed they had lab adapted at population sizes in hundreds range). What impact could this have on the measured effects? Some exploration of this issue might be useful in the discussion.
2. One of the challenging things about these types of experiments is that, because thermal performance curves are nonlinear with rapid declines at high temperatures (Sinclair et al. 2016 Ecol. Lett.), we would of course expect selection to be stronger in the heat. But if that's the case, then is this a fair test? My sense is that the major headline result of the divergence between genotypic vs phenotypic repeatability is in fact strengthened by this, but I would like to see a defense of this design in the discussion, and ideally, some information about the underlying thermal performance curve of the populations. In particular, since populations often spend time *below* their thermal optimum (Martin and Huey 2008 Am. Nat.), we might imagine that cold adaptation could be under stronger selection in the wild than was measured here.
3. Figure 2: I would prefer to see figures where the assay temperatures are matched—showing each line at its local temperature regime confounds the direct effect of temperature with the effect of adaptation.
4. Gene Ontology terms can often be unsatisfying and vague, and it can be hard to find much meaning from them. Have the authors considered using KEGG (with the KEGGREST package in R), which would map their particular changes to metabolic pathways? It would be interesting to see where pathway enrichment falls—does it look more like the changes in phenotype or genotype?

A few minor points:

1. I would recommend the authors review and cite the most updated version of Biochemical Adaptation, which was published in 2017.
2. I don't think I would characterize 23 degC as a thermal “extreme” on the cold end. I would probably recommend softening some of the language that does so.

*****END*****

Version 1:

Decision Letter:

7th February 2025

Dear Alexandre,

I hope you're doing well.

Thank you for submitting your revised manuscript "Strong Selection, but low repeatability: Temperature-specific effects on genomic predictions of adaptation" (NATECOLEVOL-24092419A). It has now been seen again by the original reviewers

and their comments are below. The reviewers find that the paper has improved in revision, and therefore we'll be happy in principle to publish it in Nature Ecology & Evolution, pending minor revisions to satisfy the reviewers' final requests and to comply with our editorial and formatting guidelines.

We will now perform detailed checks on your paper and will send you a checklist detailing our editorial and formatting requirements in about a week. Please do not upload the final materials and make any revisions until you receive this additional information from us.

[redacted]

Reviewer #1 (Remarks to the Author):

The revised manuscript is a strengthened version of an already strong piece of work. All of my issues and concerns have been addressed.

Reviewer #2 (Remarks to the Author):

The authors have taken the effort to address my main concerns, including several additional analyses. It was reassuring to see that the qualitative conclusions from the parallelism analysis were robust to SNP inclusion criteria. I have no further significant suggestions or concerns, and think the paper will make a nice contribution to the literature on both parallel evolution and thermal adaptation

Reviewer #1 (Remarks to the Author):

Given the historical nature and role of random and stochastic processes in evolution, the predictability of evolutionary outcomes has been a topic of major interest in recent decades. The evidence largely suggests that contingency and predictability of outcomes varies depending on the organisms being examined and the conditions under which evolution takes place, and altogether indicates that much more study is needed to identify important factors involved. One suggestion from theory is that strong selection promotes predictability. This manuscript describes research designed to examine this suggestion. It specifically looks at the predictability of adaptation of seed beetles to elevated or depressed temperature during experimental evolution, with analysis done at the levels of genome sequence change to life history phenotypes. They hypothesized that elevated temperatures, as is seen with global warming, exerts a stronger selection than reduced temperatures, and that so they predicted seeing more predictable evolution in the elevated temperature conditions. They found that phenotypic evolution under elevated heat is more predictable, more parallel, and faster than under depressed ones, but that genotypic evolution was the reverse. These findings were contrary to what they had expected, and they conclude that they are owing to epistatic interactions that reduce the ability to predict phenotypic adaptation from genotypic data. Somewhat surprisingly given earlier work in microbial systems, they found that antagonistic pleiotropy was highly involved in thermal adaptation, which was interesting.

I found this manuscript to be clear, well-written, and compelling. The research it describes is very impressive, deep, and incisive. The findings are in line with previous work that has shown similar disparities in predictability of phenotypic adaptation versus lack of predictability in genotypic changes underlying that adaptation. The explanation they give that differences between the populations evolved under the different temperature regimes as owing to epistasis is compelling and well supported. It is in line with ongoing findings that epistasis is extremely important in adaptation, and plays strong roles in evolutionary trajectories during selection. Moreover, they found important differences in repeatability across genetic backgrounds, and, I think, rightfully conclude that this suggests a strong role for historical contingency. Indeed, this finding is in line with the emerging consensus that genetic relationships are very important to predictability of evolution across lineages. Unfortunately, as the authors note (and I agree with them), this means that it can be difficult to usefully generalize expected evolutionary patterns from limited numbers of study systems, which is problematic for developing models of how organisms will evolve in response to climate change.

To be honest, after going through the paper multiple times, I do not have anything to really call out or criticize. This is a superb piece of work. The mathematical and computational analyses are beyond my understanding, so I hope the other reviewer(s) can identify any defects. The experiments were well-conducted and clearly described, with issues such as variation in generations under the two conditions clearly noted and accounted for. The conceptual analysis

is spot on, rigorous, and not overblown. It says a lot that, every time I had a question about some aspect of the work during my initial read through, I found it soon addressed later in the text. This paper shows a great deal of deep thought and careful consideration. Usually, I am able to at least point out bad grammar or other such that should be corrected, but, if such issues are present, I did not notice them. This is one of the rare instances in which I do not feel that I can make any suggestions that would help to improve the paper. This is an excellent manuscript that describes excellent and important work in this area, and I can see it becoming a well-cited part of the literature on evolutionary predictability and contingency.

We would like to thank the reviewer for their very thorough and accurate summary of our work and we are greatly appreciative of the positive feedback and happy to see that they share our excitement about this work.

Reviewer #2 (Remarks to the Author):

This paper combines phenotypic and genomic data from an experimental evolution design in beetles to assess how repeatable (or “parallel”) evolutionary change is when adapting to hot and cold environments. They find that phenotypic change is more parallel when adapting to hot vs cold environments, but the genomic data indicates the opposite. Based on this and some other results the authors suggest a potentially more important role for pleiotropy in adaptation to hot environments. This paper is interesting, and the dataset, analyses and set of questions addressed is impressive and certainly within the scope of a journal like NEE. However, I do have a few issues with how the analysis of repeatability was performed that I think need to be addressed before I will fully endorse the authors story here.

We are very grateful for this positive assessment of our study and also for the thoughtful and detailed constructive feedback given below. We have made an effort to accommodate these suggestions in our manuscript (see comments below).

A fundamental analytical challenge with addressing these questions about repeatability of evolution is that it is very hard to interpret what an angle in high dimensional trait space means. Put simply, when we study a lot of traits at once and examine angles between evolutionary change vectors from replicate lineages evolving to a common gradient, we have the following problems: 1) it is extremely hard to intuitively interpret angles in any meaningful way (they are not an effect size), and if we examine many lineages and many traits, interpreting individual pairwise angles can fail to fully reveal clustering of vectors in certain directions of trait space, 2) it does not make sense to compare, even informally, two (or more) angles if the trait dimensionality between the two vector pairs differs, the angles simply are not comparable. The authors are clearly aware of these issues, but I don't think they have been adequately addressed.

For example, the main counterintuitive result of the paper is that angles for genomic data are more orthogonal in hot than cold, opposite the phenotype data. Yet, the authors found on average about twice as many SNPs under selection in Cold lines than in Hot lines (based on the

supplemental table), but all of these SNPs were included in this analysis of angles. That is, in this analysis there were many more SNPs that showed no evidence of being under selection (or even more disturbingly, were even polymorphic in all of the ancestral populations!!, see below) in hot vs cold. So we know that most of the SNPs showed no change in the hot lines, and when we include all of these SNPs sure enough we find that evolution was more repeatable in the cold lines. Put simply I think the result is purely an outcome of the fact that more SNPs were under selection in the cold lines, but all the SNPs were included in the analysis. One could make a philosophical argument that it would be valid to include all of the SNPs anyways, but given the authors are specifically filtering by those under selection and there is a huge discrepancy in the number under selection in each environment, this creates a problem for me with how we interpret these results.

We would first like to thank the reviewer for this comment, which is very accurate in describing a potential problem with the analysis. We do, however, note that some of our previous analyses (e.g. on gene re-use; upset plots) showed that repeatability across backgrounds is lower for heat-adaptation also when only considering regime-specific SNPs. For the analyses of genomic angles, we included all selected SNPs for analyses of hot and cold lines in order for the dimensionality to be the same, to make the angles comparable in that respect (i.e., with reference to the other perspective on dimensionality brought up by the reviewer). However, we agree with the reviewer that this strategy can potentially cause bias by including more neutral SNPs in the hot lines than in the cold lines (leading to apparent lower repeatability in hot lines).

To explore if this potential artefact was driving our results, we followed the reviewer's suggestion to include SNPs selected specifically in the regime for which the angles were calculated (i.e. candidate SNPs detected for adaptation to heat only when analyzing angles between hot populations). Additionally, we performed an analysis where we included only SNPs that were shared among all three genetic backgrounds (as a response to another of the reviewer's comments - see below). These analyses showed that the qualitative results hold irrespective of which SNP-set we use. We have included the results of all these analyses as additional Supplementary material, and have clarified these important issues in the main text (Lines 230-233, Figs S8 & S11). Again, we would like to thank the reviewer for these comments, which we think has helped improve the quality of our manuscript further.

The alternative to the above approach would be to filter SNPs separately by environment, and analyze parallelism in change separately for hot and cold, using the hot and cold SNPs. Of course then there would be different dimensionality and so the angles in the “hot” analysis could not be directly comparable to the “cold”, but nonetheless there are several standardized effect sizes (Watanabe 2022) that could be presented to assess relative parallelism in adaptation to each environment. I think this would be preferable to the present approach, and could likely change the story entirely: if we are just focusing on the SNPs under selection in each environment, is evolution still more parallel in cold?

We were content to see that the qualitative results hold irrespective of the SNP-set used (see above).

Second, I don’t think the authors have been clear enough about how we really can’t contrast the angles from the trait data with those from the genomic data. They have 7 phenotypic traits and over 400,000 genomic traits. The authors have done some simulations so are aware of the null expectations, I suggest they plot the nulls to see for themselves how even only the slightest departure from pure orthogonality carries great meaning when we have 400,000 traits; I doubt the Cis on the null expectation would even be visible when plotted on Figure 3E. Put simply the value of an angle from one of the trait vector correlations simply is not relevant when we are thinking about angles from the genomic comparisons. This point about dimensionality and interpretation of angles has been made a few times recently in the literature, but it can’t be overemphasized. I actually think the genomic data might be more meaningfully “parallel” than the trait data, but it is hard to say just based on the analyses presented. All of these points should be made more obvious to readers and anyone looking at the figures.

Our aim was never to compare the angles between phenotypic data and genomic data directly, and we are sorry if this came across as our intention. Instead, our main focus was to compare the relative differences in the repeatability of adaptation to hot and cold temperature at the phenotype vs. genomic scale. We certainly agree with all the points that the reviewer makes here, and acknowledge that we probably could have done a better job at making this point more clear. Accordingly, we have made a clarifying statement about the importance of dimensionality when interpreting the reported angles (Lines 220-225) and have also added a supplementary figure showing 95% CIs for the angles under the null hypothesis together with our observed mean estimates of angles within and between backgrounds, as suggested (Supplemental figure S8).

Aside from the above issues, I think the assessment of genomic parallelism in particular is a bit simplistic. For example, Whiting et al 2022 (<https://doi.org/10.1111/2041-210X.13952>) extended De Lisle and Bolnick's eigen analysis approach to understand more complex patterns of parallel change in SNP data. Given there are 400000 SNPs here, I wouldn't be surprised if the patterns of clustering of these vectors are more complex than can be assessed by inspection of the angles.

As suggested, we ran AF-vapeR on our data to detect genomic windows of parallelism. We have presented these analyses as a new Supplementary S6. In short, this new set of analyses confirm the overall conclusion that adaptation is highly polygenic. It also shows again that selection is stronger at high heat and less parallel between backgrounds.. We highlight patterns of parallelism and anti-parallelism in a few of the key outlier genomic regions. We hope the reviewer appreciates that we have to limit the scope of our analyses here and cannot continue to expand even more on these analyses.

Finally, I struggled a bit to understand the point of the “fitness offset” analysis. It seems to be simply trying to correlate distance in a fitness measure (which is actually more of a life history measure than fitness, see below) with SNP distance. There was little conceptual or theoretical motivation provided for this, which is disconcerting especially considering that the measure of fitness is not necessarily even close to what most evolutionary biologists think of as fitness.

We aimed to assess if genomic changes can predict the level of adaptation during our experiment, and hence, we aimed to derive a fitness estimate (which we now call “laboratory fitness”) capturing this adaptation. We believe it is a good estimate of the level of adaptation in these beetle lines; the measure captures the number of adult offspring produced by a male-female mating couple at each assay temperature. Because we counted the number of adult offspring produced, this estimate also captures differences in (temperature-specific) juvenile mortality. We then took this measure and divided it by the egg-to-adult development time (a very good approximation of generation time in seed beetles), to arrive at a measure that corresponds to population intrinsic rate of growth at each temperature. We have added some additional information to the text to highlight this better (Lines 524-535).

More importantly, we do not think our presented analyses of offsets at 35C are dependent on the exact phenotypic measures used, noting that the results are qualitatively the same using the total phenotypic distance (instead of our estimate of lab fitness).

Specific/minor comments

I suggest the authors try to reduce acronym use. We have PH, PC, SP, AP, AW, WL and more. Just write out the words, please, I had to think about what these meant each time I came across them

Agreed - we have done so in the main text.

Line 178 A very minor point: Treating this type of data as a correlation matrix was actually proposed by De Lisle and Bolnick (Ref 98), Watanabe proposed some different ways to analyze the correlation matrix

Corrected - thank you!

Line 200-226 This description was a bit hard to follow. I think the authors are comparing pairwise distances in 7-dimensional trait space, but it could be a trait-by-trait analysis as the description is unclear.

It's correct, it's pairwise distances, we have tried to clarify this in this section (Line 623-624).

Line 336 I have never seen such a measure of fitness, and this sounds more like some description of life history than of fitness itself, as it will measure variation in how resources are allocated to offspring production through time (i.e., life history). Lifetime reproductive success is what most people think of as fitness, and this can be multiplied by a juvenile survival estimate if that is available.

This is very close to that - lifetime adult offspring production, including juvenile survival, divided by egg-to-adult development time as an approximation of generation time at each temperature (see above).

Line 370-379 This is also a nice demonstration of why it might often be useful to think of parallelism and convergence as separate but related phenomenon

Agreed!

Line 467 I think "partly" is a severe understatement; you literally have hundreds of thousands of SNPs, and just 7 phenotypic traits. See above comment

We agree and have now clarified this.

Line 504 Wait, why wasn't the main analysis based on this subset to begin with? It makes no sense to me to assess parallelism across a set of SNPs that weren't segregating in some of the ancestral populations, since unless there is significant mutational input (doubtful in a relatively short timespan in an experiment with small N_e) then we wouldn't expect to see any change at these loci anyways!!

The probability of observing repeated evolution has often been discussed with reference to different factors, and one of those factors is the occurrence of ancestral polymorphisms shared (or not) between different lineages. Our approach was motivated by this framework and we wanted to explore the effect of different factors. This is why we started off by comparing populations using all SNPs, and then we restricted analyses to those SNPs that were shared among all populations, expecting repeatability to be strongly increased if ancestral polymorphism (or lack thereof) was an important factor in our experiment. We repeated analyses on the level of both SNPs and genes (and GO terms) expecting more repeatability at the highest level of organization (phenotypes > genes > SNPs). We find support for this later prediction. However, ancestral polymorphism, although increasing repeatability slightly overall, did not explain differences in genomic repeatability between temperatures. We have added an explanation of our rationale in the Results (lines 206-217).

Reviewer #2 (Remarks on code availability):

I downloaded the scripts and data, and didn't run the whole thing but did check some of the matrix calculations. It does appear all the data and code are presented, and it is pretty clear and easy to interpret their scripts. I think they have done a fine job on that

Reviewer #3 (Remarks to the Author):

In this manuscript, the authors examine the repeatability of evolution to high and low temperatures in a seed beetle. They find that phenotypic evolution was faster and more repeatable in the heat, but that the opposite was true for genotypic change, and that there was a strong effect of genetic background. I found the manuscript generally clear and easy to read, and I commend the authors for their thorough investigation.

1. One challenge I have with this manuscript is the relatively low population size. Insect population sizes can be in the millions, and yet here the population size is 215 (and indeed they had lab adapted at population sizes in hundreds range). What impact could this have on the measured effects? Some exploration of this issue might be useful in the discussion.

We agree that this is an important point. Indeed, our effective population sizes ($N_e \sim 200$) and census size ($N = 600$) are indeed much lower than typical for many insect populations, but perhaps quite relevant for many of the populations targeted by conservation genomic approaches (e.g. analysis of genomic offsets).

Our relatively low population size will surely have limited our possibility to detect weak selection of alleles of small effect (selection coefficients < 0.005), but we note that this is true for most studies on experimental evolution, except for those focused on microorganisms. The effect of genetic drift at those sites will have led to low repeatability. However, we focus our analyses on those SNPs with selection coefficients with $s > 0.01$, and for those we would expect high

repeatability in the absence of genetic redundancy (genetic adaptation being highly polygenic) and epistasis. Yet, we find the very strong effect of genetic background and temperature-specific effects, even when focusing on shared polymorphisms only, so we doubt the main conclusions about repeatability are affected by our effective population size being lower than that in nature. We agree that this is an important point that we now address in more detail in the Discussion (Lines 443-459).

2. One of the challenging things about these types of experiments is that, because thermal performance curves are nonlinear with rapid declines at high temperatures (Sinclair et al. 2016 Ecol. Lett.), we would of course expect selection to be stronger in the heat. But if that's the case, then is this a fair test? My sense is that the major headline result of the divergence between genotypic vs phenotypic repeatability is in fact strengthened by this, but I would like to see a defense of this design in the discussion, and ideally, some information about the underlying thermal performance curve of the populations.

We agree that this is a clear expectation regarding the repeatability of phenotypic responses, which is why we designed our experiment in the way we did. We were also content to see this expectation fulfilled in the empirical data on phenotypes. To strengthen this argument further, we have included thermal reaction norms for laboratory fitness (same estimate as that used for calculating offsets) of the three ancestors (added as a new panel to Figure 1; and see methods for "Ancestral Thermal Performance"). This shows the typical asymmetric shape of the reaction norms, with a steeper fitness gradient at 35C relative to 23C. We also note that we cannot maintain *C. maculatus* at 37C in the lab (Berger & Liljestrand-Rönn 2024 Ecol Lett), so 35C is close to the thermal maximum. We hope this contributes to further clarity regarding choice of experimental temperatures.

The question whether selection is stronger at hot vs cold temperature across the entire genome is not as straight-forward. For example, environments that are more stressful do not lead to stronger selection in general terms (e.g. Agrawal & Whitlock 2010 TREE). Nevertheless, for hot temperature, theory suggests that this might be the case, because temperature affects a large number of genes across the genome (potentially all protein coding genes (see e.g. Berger et al. 2021 Proc B, and Aggozini and Dill 2019 PNAS). This was also our motivation for choosing temperatures equidistant from the benign ancestral temperature; expecting more severe effects of a shift towards hot, relative to cold, temperature. We found support for stronger selection both at the phenotypic and genomic level, but repeatability did not follow the same pattern and expectation - which is the focus of our study. We note that we already discuss the biophysical basis of thermal adaptation at length in the manuscript in several places (e.g. Supplementary on temperature-specific epistasis), and are well beyond the word limit with the current version. We have therefore not incorporated even more discussion on this topic.

In particular, since populations often spend time *below* their thermal optimum (Martin and Huey 2008 Am. Nat.), we might imagine that cold adaptation could be under stronger selection in the wild than was measured here.

We agree with this, especially for temperature populations, but perhaps not for tropical species like *C. maculatus* (see e.g. Baur et al. 2024 Evol Lett). Here we compared repeatability of adaptation at hot and cold temperature with the same time spent at each temperature, which we think is the most fair and direct comparison. Of course, if a species does spend less time at a particular temperature, alleles with effects limited to this particular temperature will experience relaxed selection and have their fate largely decided by genetic drift, leading to low repeatability.

While we think this is an interesting discussion, we have limited space to expand on this comment as we are already beyond the word limit.

3. Figure 2: I would prefer to see figures where the assay temperatures are matched—showing each line at its local temperature regime confounds the direct effect of temperature with the effect of adaptation.

When we introduce the analyses of phenotypic data, we reference our other publication where the phenotypic data are displayed as the reviewer requests (Burc. et al. 2024, bioRxiv). Note that temperature-specific adaptation in multivariate space is also visible in Fig. 1 (Phenotypic results of the PCAs at each assay temperature). We chose the current figure on phenotypic data as we felt it made it easier to evaluate parallel divergence from the ancestor between the replicate evolution lines for individual traits.

4. Gene Ontology terms can often be unsatisfying and vague, and it can be hard to find much meaning from them. Have the authors considered using KEGG (with the KEGGREST package in R), which would map their particular changes to metabolic pathways? It would be interesting to see where pathway enrichment falls—does it look more like the changes in phenotype or genotype?

This is a very interesting and constructive suggestion. We did try this but unfortunately the annotation of the *C. maculatus* genome is still not good enough for this analysis to provide insights as those for other model species like *Drosophila*. We therefore were not able to include this analysis to the manuscript.

A few minor points:

1. I would recommend the authors review and cite the most updated version of *Biochemical Adaptation*, which was published in 2017.

Done - thanks!

2. I don't think I would characterize 23 degC as a thermal "extreme" on the cold end. I would probably recommend softening some of the language that does so.

We agree - we have changed wording. (note that these populations cannot complete their life cycle at 17C, and 23C is still suboptimal in terms of development, but OK for fecundity)

Thank you for the constructive comments!

Response to Reviewers

Reviewer #1:

Remarks to the Author:

The revised manuscript is a strengthened version of an already strong piece of work. All of my issues and concerns have been addressed.

Reviewer #2:

Remarks to the Author:

The authors have taken the effort to address my main concerns, including several additional analyses. It was reassuring to see that the qualitative conclusions from the parallelism analysis were robust to SNP inclusion criteria. I have no further significant suggestions or concerns, and think the paper will make a nice contribution to the literature on both parallel evolution and thermal adaptation

We thank both reviewers for their time and effort in reviewing our manuscript. We agree that their suggestions have improved the quality of our work and have strengthened our results and conclusions further.